# Challenges of Generating Structurally Diverse Graphs

**Fedor Velikonivtsev**
HSE University, Yandex Research
fvelikon@yandex-team.ru

**Mikhail Mironov**
Yandex Research
mironov.m.k@gmail.com

**Liudmila Prokhorenkova**
Yandex Research
ostroumova-la@yandex-team.ru

## Abstract

For many graph-related problems, it can be essential to have a set of structurally diverse graphs. For instance, such graphs can be used for testing graph algorithms or their neural approximations. However, to the best of our knowledge, the problem of generating structurally diverse graphs has not been explored in the literature. In this paper, we fill this gap. First, we discuss how to define diversity for a set of graphs, why this task is non-trivial, and how one can choose a proper diversity measure. Then, for a given diversity measure, we propose and compare several algorithms optimizing it: we consider approaches based on standard random graph models, local graph optimization, genetic algorithms, and neural generative models. We show that it is possible to significantly improve diversity over basic random graph generators. Additionally, our analysis of generated graphs allows us to better understand the properties of graph distances: depending on which diversity measure is used for optimization, the obtained graphs may possess very different structural properties which gives a better understanding of the graph distance underlying the diversity measure.

## 1 Introduction

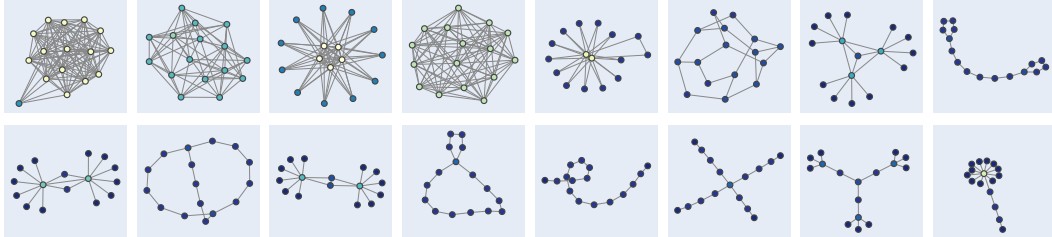

Figure 1: A sample of generated graphs

Many real-world objects can be naturally represented as graphs: biological and chemical entities (atoms, molecules, proteins, metabolic maps), interaction networks (social and citation networks, financial transactions), road maps, epidemic spreads, and so on. That is why the analysis of graph-structured data is an important and rapidly developing research area.

To generate realistic graph structures, many random graph models have been proposed in the literature (Boccaletti et al., 2006). Such models aim to imitate properties typically observed in natural structures: power-law degree distribution, small diameter, community structure, and others. Each

38th Conference on Neural Information Processing Systems (NeurIPS 2024).

random graph model captures some of these properties; thus, the generated graphs are inevitably similar in certain aspects.

On the other hand, for some applications, it is important to be able to generate a set of graphs that are *structurally diverse*. For instance, if one needs to automatically verify the correctness of a graph algorithm, estimate how well a heuristic algorithm approximates the true solution for a graph problem, or evaluate neural approximations of graph algorithms (Veličković and Blundell, 2021). In all these cases, algorithms and models should be tested on as diverse graph instances as possible since otherwise the results can be biased towards particular properties of the test set (Georgiev et al., 2023). In other words, we need representative graphs that 'cover' (in some sense) the space of all graphs. Datasets consisting of diverse graphs can also be useful for evaluating graph neural networks and their expressive power. In this direction, Palowitch et al. (2022) propose the GraphWorld benchmark that consists of graphs with various statistical properties. An important part of the benchmark is the relation between graph structure and node labels: graphs with different homophily levels can be constructed. However, generated graph structures are limited to the degree-corrected stochastic block model (Karrer and Newman, 2011) and thus do not cover all complex connection patterns that graphs may have.

To the best of our knowledge, the problem of generating a dataset where graphs are maximally diverse has not been addressed in the literature yet. In this paper, we fill this gap. For this purpose, we first need to define diversity of a set of graphs which is already a challenging task. To measure diversity, we define dissimilarity (distance) for a pair of graphs and then aggregate the pairwise graph distances into the overall diversity measure. In this regard, we show that popular methods of measuring diversity have significant drawbacks and suggest using Energy since it satisfies certain important desirable properties.

After we have defined a performance measure for our problem, several approaches can be used to optimize it. We develop and analyze the following strategies: a greedy method based on diverse random graph generators, a local graph optimization approach, an adaptation of the genetic algorithm to our problem, and a method based on neural generative modeling. For the simplest greedy algorithm, we provide theoretical guarantees on the diversity of the obtained set of graphs relative to the maximal achievable diversity (for a given pre-generated set of graphs to choose from).

We empirically investigate the proposed strategies and show that it is possible to significantly improve the diversity of the generated graphs compared to basic random graph models. In addition to the numerical investigation of diversity measures, we also analyze the distribution of graph structural characteristics and relations between them. Here we also observe significantly improved diversity. Moreover, since we consider diversity measures based on several graph distances, our results shed light on the properties of these graph distances. Indeed, depending on the function we optimize, the structural properties of the generated graphs can vary since graph distances focus on different aspects of graph dissimilarity. Thus, by inspecting the properties of generated graphs, one can better understand what graph characteristics a particular graph distance is sensitive to.

In summary, this work formulates and investigates the problem of generating structurally diverse graphs that can serve as representative instances for various graph-related tasks. Still, many challenges remain to be solved and we hope that our work will encourage further theoretical and practical research in this field.

## 2 Defining diversity for a set of graphs

### 2.1 Problem setup

As discussed above, for various graph-related problems, it can be essential to have representative graphs that are structurally diverse. Intuitively, such graphs are expected to cover (in some sense) the space of all graphs.[1] This section discusses how to define diversity and why it is non-trivial.

Let us start with a motivating example. The most basic random graph generator is the Erdős-Rényi model. In this model, an edge between any two nodes is added with probability $p$ independently of other edges. If $n$ is fixed and $p = 0.5$, then every simple graph on $n$ nodes can be generated equally

---

[1]In this work, we use the terms 'diversity' and 'coverage' interchangeably.

likely (assuming that the nodes are enumerated), and thus one may think that this model generates representative graphs.

However, it is known that with high probability graphs generated according to the Erdős-Rényi model have typical properties and thus are all very similar to each other with high probability (Erdős et al., 1960). For instance, as illustrated in Figure 2, for the Erdős-Rényi model with $p = 0.5$ (ER-0.5), the average node degree and average clustering coefficient are concentrated near $(n-1)/2$ and 0.5, respectively. Varying $p$ (ER-mix) allows one to get all possible values of these characteristics, but they are linearly dependent in expectation. Thus, the space of all possible combinations of characteristics cannot be covered by the Erdős-Rényi model.

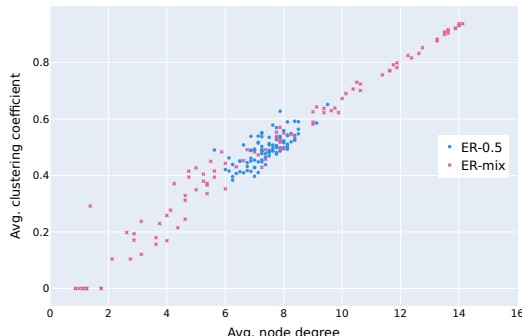

Figure 2: Average node degree and average clustering coefficient in the Erdős-Rényi model with $n = 16$ and $p = 0.5$ (ER-0.5) or varying $p$ (ER-mix)

Intuitively, by *diverse graphs* we mean those having different structural properties such as degree distribution, pairwise distances, subgraph counts, and so on. However, this intuition is hard to formalize as one may potentially come up with infinitely many properties. On the other hand, defining graph dissimilarity is closely related to *graph distances*. Graph distances have been studied for a long time, and many variants capturing different graph properties exist in the literature (Tantardini et al., 2019). We review some of them in Section 2.2.

Now, assume that we have a multiset of $N$ graphs $S = G_1, \ldots, G_N$. Throughout the paper, we consider undirected graphs without self-loops and multiple edges. Assume that we are also given a distance measure $D(G, G')$ that evaluates dissimilarity between two graphs. Then, we define diversity as:

$$\text{Diversity}(S) = F(\{D(G, G') : G, G' \in S\}), \tag{1}$$

where $F$ is some function that computes diversity given a set of pairwise distances. The choice of $F$ is also non-trivial, and we discuss possible approaches in Section 2.3.

After we have defined the distance $D(\cdot, \cdot)$ and the measure of diversity, our primary goal is to find a multiset of graphs $\bar{S}$ of size $N$ to maximize its diversity:

$$\bar{S} = \underset{G_1, G_2, \ldots, G_N \in \mathcal{G}_n}{\arg\max} \text{Diversity}(\{G_1, G_2, \ldots, G_N\}), \tag{2}$$

where $\mathcal{G}_n$ is the set of all graphs with $n$ nodes.[2]

## 2.2   Graph distances

This section discusses how one can define distance between two graphs. As mentioned above, this task is highly non-trivial, and the literature on this topic is abundant (Tantardini et al., 2019; Hartle et al., 2020).

Some graph distance measures are based on the optimal node matching between two graphs: first, the correspondence between nodes is found, and then some distance between two adjacency matrices can be computed (a popular example of this type is *graph edit distance*). However, this approach can only be applied to graphs having the same number of nodes (to achieve this in general case, one may add zero-degree nodes to the smaller graph). Also, finding the optimal matching between nodes is usually computationally expensive.

Another class of distance measures is based on computing some descriptor (vector representation) for a graph and then measuring the distance between two graph representations. Such measures usually violate the positivity axiom of a metric space since the distance between two different graphs can be

---

[2]For simplicity, we assume that the number of nodes $n$ is the same for all graphs. Note that $n$ can naturally be considered as an upper bound on the number of nodes: smaller graphs can be obtained if some of the nodes have zero degrees.

equal to zero. Indeed, if we guarantee that $D(G, G') = 0$ if and only if $G$ and $G'$ are isomorphic, then computing the distance is at least as hard as graph isomorphism testing, which is infeasible for most applications. Thus, when computing a graph representation, we inevitably lose some information about the graph. Various approaches for creating graph descriptors exist in the literature. Some of them use the spectrum of the graph Laplacian (or its normalized variant) that is known to encode some important structural information (Ipsen and Mikhailov, 2002; Wilson et al., 2005; Tsitsulin et al., 2018). Other approaches are based on local statistics, such as graphlets (Yaveroğlu et al., 2014). Each graph distance captures some properties of a graph and can be insensitive to others. We refer to comparative studies of graph distances (Tantardini et al., 2019; Hartle et al., 2020) for a more comprehensive list of known measures and the analysis of their properties. In more recent work, Thompson et al. (2022) suggested graph representations based on an untrained random GNN. Such representations can also be used for computing graph distances.

Our paper does not aim to answer which graph distance is better. Each distance captures particular graph properties and the resulting set of generated diverse graphs may significantly depend on this choice. In our experiments, we consider several representative options. As a result, our analysis of generated graphs gives some additional insights into the properties of graph distances.

### 2.3 Measuring diversity for a set of elements

In this section, we discuss the problem of measuring diversity for a set of elements given their pairwise distances (or similarity values). This problem was addressed in several recent papers (Xie et al., 2023; Friedman and Dieng, 2023) discussed in more detail in Appendix A.1. However, as we show, none of the proposed approaches are fully suitable for our task.

Probably the most natural and widely-used measure for quantifying diversity is the *average pairwise distance* between the elements. However, we note that this measure is not suitable in our case since optimizing it may lead to degenerate configurations. For instance, consider a toy experiment with dots distributed on a line segment. As shown in Figure 3a, optimizing the average pairwise distance forces many points to collapse into one (at the endpoints of the line segment), which is clearly not a desirable behavior. This happens since Average does not take into account whether the elements of a dataset are unique or well isolated from each other.

Another possible measure that does take uniqueness of elements into account is the *minimum pairwise distance* (often referred to as *Bottleneck* in the literature). However, this measure is not sensitive to all the distances but the minimal one and thus cannot distinguish vastly different configurations.

Motivated by these examples, we formulate two properties that a good diversity measure is expected to satisfy. We assume that we are given a multiset $S$ of $N$ elements and for each pair of elements $G, G' \in S$ we know the distance $D(G, G')$ between them. Note that we are interested in maximizing diversity for a *fixed* number of elements $N$, which simplifies the requirements since we do not have to deal with how diversity changes when the number of elements increases.

**Monotonicity** Suppose we are given two multisets $S, S'$ both consisting of $N$ different elements and a bijection $g : S \to S'$ between them. Assume that for any $G_i, G_j \in S$ we have

$$D(G_i, G_j) \leq D(g(G_i), g(G_j)) \tag{3}$$

with strict inequality for at least one pair $i, j$. Then we require $\text{Diversity}(S) < \text{Diversity}(S')$.

This property describes the essence of diversity measures: larger pairwise distances should lead to higher diversity values. Thus, a good measure of diversity should be monotone.

**Uniqueness** Suppose $S$ consists of $N$ different elements $G_1, \ldots, G_N$. Denote by $S_{ij}$ the multiset obtained from $S$ by removing $G_i$ and adding the second copy of $G_j$ for $j \neq i$, that is $S_{ij} := (S \backslash \{G_i\}) \cup \{G_j\}$ Then, we require $\text{Diversity}(S) > \text{Diversity}(S_{ij})$.

In other words, this property requires that given $N - 1$ elements, adding a unique element results in a higher diversity than duplicating an already existing element. This property is very intuitive since to increase the coverage it is clearly better to add a new element than to duplicate an existing one.

Note that the average pairwise distance does not have the uniqueness property, thus optimizing it may lead to degenerate solutions. Clearly, the minimum pairwise distance does not have monotonicity.

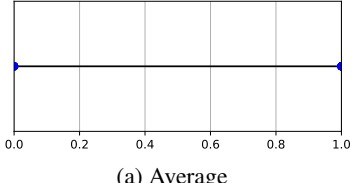
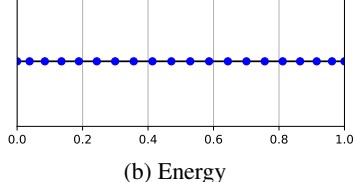

(a) Average                                       (b) Energy

Figure 3: Optimized Average or Energy on a line segment

Moreover, it turns out that none of the measures from (Xie et al., 2023; Friedman and Dieng, 2023) has both these properties, see Appendix A.2 for the details.

Thus, we propose an alternative diversity measure motivated by the *energy of a system of equally charged particles*. Namely, given a constant $\gamma > 0$, we define the Energy of a set of graphs $S$ as

$$-\frac{1}{N(N-1)} \sum_{i \neq j} \frac{1}{D(G_i, G_j)^\gamma}. \tag{4}$$

The parameter $\gamma$ affects how strongly we penalize small pairwise distances. In an extreme case of $\gamma \to \infty$, this measure becomes equivalent to Bottleneck. All our theoretical results hold for any $\gamma > 0$.[3] For our experiments, we use $\gamma = 1$, so (4) can be naturally interpreted as the average pairwise energy for a system of equally charged particles (we multiply by -1 to get a measure that is larger for more diverse sets of graphs).

Our toy example in Figure 3b shows that when being optimized Energy leads to a diverse configuration, in contrast to Average. Regarding our formal properties, monotonicity is obviously satisfied by Energy (4). To show uniqueness, we note that any multiset with pairwise different elements has some finite negative diversity and any multiset with two copies of one element has diversity $-\infty$.

**Proposition 2.1.** *Energy* (4) *satisfies both monotonicity and uniqueness.*

Despite having these desirable properties, Energy still has a shortcoming: it can be unboundedly large when two elements become too close to each other. However, there are currently no better alternatives, as shown in a recent paper by Mironov and Prokhorenkova (2024) that extends the analysis of diversity measures in terms of the desirable properties they satisfy. We refer to Appendix A.4 for a discussion. In our experiments, we use Energy (combined with several graph distances) as our primary measure of diversity and also consider Average as an additional measure.

## 3 Algorithms for diversity optimization

In the previous section, we discussed how to measure diversity and why this task is non-trivial. In this section, we propose several approaches for diversity optimization. Our goal is to investigate diverse algorithms: from a basic approach based on random graph generators to a more advanced one based on neural generative modeling.

Our algorithms can be applied to arbitrary diversity measures. However, for scalability purposes, we restrict ourselves to measures that can be written in the following way. Suppose we are given a set of size $N$ and any element $G$ from this set. Denote the subset of all elements excluding $G$ by $S = \{G_1, G_2, \ldots, G_{N-1}\}$. Then, the diversity of the original set can be written as:

$$\text{Diversity}(\{G\} \cup S) = g(f(G, S), c(S)), \tag{5}$$

where $g$ is a function that is monotone w.r.t. both arguments, $f(G, S)$ depends only on the distances $\{D(G, G_i) : G_i \in S\}$, and $c(S)$ is a value that depends only on $S$ (and does not depend on $G$). We call such function $f(G, S)$ a *fitness* of a graph $G$ w.r.t. a set of graphs $S$. The measures considered in this study satisfy (5). For instance, for Energy, $g$ can be the sum, $f(G, S) = \frac{1}{N(N-1)/2} \sum_{G_i \in S} \frac{1}{D(G, G_i)^\gamma}$, and $c(S) = \frac{1}{N(N-1)/2} \sum_{i<j} \frac{1}{D(G_i, G_j)^\gamma}$. For Bottleneck, $g$ can be the minimum, $f(G, S) = \min_{G_i \in S} D(G, G_i)$, and $c(S) = \min_{i<j} D(G_i, G_j)$. Note that computing the fitness $f(G, S)$ requires $N - 1$ distance computations.

---

[3]We illustrate the effect of $\gamma$ on the obtained diverse configurations in Appendix A.3.

It is important to note that standard machine learning approaches cannot be directly applied to our task: usually, generative algorithms require a training set that they try to imitate. In our case, there is no training set since the aim is to generate graphs that are maximally dissimilar to each other.

## 3.1   Greedy algorithm

The main idea of this algorithm is to build a set of diverse graphs iteratively by adding at each step the most suitable graph from a predefined set $\hat{S}$ of a much larger size. This set can be either user input, the result of another algorithm, or a set of graphs generated by random graph models. The process initiates by randomly choosing a graph from $\hat{S}$. At each step, the most suitable graph from $\hat{S}$ is chosen according to the fitness $f(G, S)$, where $S$ is the currently selected set of graphs.

A detailed description of the algorithm is given in Appendix C.1. We also provide the analysis of computational complexity and a lower bound on the diversity of graphs returned by the greedy algorithm relative to the diversity of the initial set $\hat{S}$ (see Theorem C.1).

## 3.2   Genetic algorithm

The genetic algorithm enhances the diversity of a graph population through evolutionary operations. Starting with an initial set of $N$ graphs, it iteratively refines this set by selecting pairs of graphs as parents and generating a child through crossover and mutation processes. This child can replace the less-fit graph in the population if it increases the overall diversity; otherwise, the algorithm tries to find a more suitable offspring by repeating the process. To prevent itself from getting stuck in local optima, the algorithm can accept a candidate that decreases the overall diversity if the number of unsuccessful attempts exceeds a certain threshold. The algorithm iterates for a predefined number of iterations, ultimately evolving the population towards greater diversity. This approach adapts principles from genetics to solve optimization problems, as we try to preserve beneficial graph characteristics while at the same time introducing novel configurations to achieve a diverse set of graphs.

The details of this algorithm are given in Section C.3, where we also analyze the complexity of the algorithm.

## 3.3   Local optimization algorithm

The main idea of the local optimization algorithm is the refinement of the diversity of a graph population by iteratively modifying individual graphs. Starting from an initial set, we randomly sample graphs and make small modifications to their structure (single edge addition/deletion). Then, if the overall diversity improves, we accept the change. As in the other algorithms, we can accept less fit modifications after consecutive failed attempts to prevent stagnation at a local optimum.

The details of this algorithm are given in Section C.4, where we also analyze its computational complexity. Since local optimization makes small modifications at each step, this approach is expected to be most efficient when the input set of graphs is already sufficiently diverse. Thus, when we combine several algorithms, local optimization is always the last step.

## 3.4   Iterative graph generative modeling

Neural generative models are known to be a powerful tool for generating graphs that imitate a given distribution (You et al., 2018; Martinkus et al., 2022; Vignac et al., 2023). Hence, we aimed to investigate whether such approaches can be used for generating graphs that are structurally diverse. In this case, there is no predefined distribution that needs to be captured. We address this via the following iterative procedure. The process starts from an initial graph set $S_0$ and then iteratively enhances the diversity. At each iteration, the current set of graphs $S_i$ is used to train a generative model. Then, this model is used to generate a significantly larger set of new graphs. From this new set, a smaller subset of diverse graphs $S_{i+1}$ is selected via the greedy approach. We expect that $S_{i+1}$ is more diverse than $S_i$. So, we repeat the process by training a neural generative model on the new set $S_{i+1}$. For the neural network architecture, we use Discrete Denoising Diffusion Model (DiGress) (Vignac et al., 2023). We refer to Appendix C.5 for a detailed description of our approach.

# 4 Experiments

In this section, we analyze and compare the algorithms for generating diverse graphs described above. Then, we analyze generated graphs and discuss how the choice of a particular graph distance affects the structures of the obtained graphs.

**Setup**  In our experiments, we consider four representative distance measures: heat and wave NetLSD (Tsitsulin et al., 2018), Graphlet Correlation Distance (Yaveroğlu et al., 2014), and Portrait Divergence (Bagrow and Bollt, 2019). We select these distances to be diverse: NetLSD is based on the Laplacian eigenvalues (we use NetLSD-heat and NetLSD-wave variants), Graphlet Correlation Distance (GCD) uses local structures, while Portrait Divergence (Portrait-div) takes into account both local and global properties. A detailed description of these measures is given in Appendix B.

Following Section 2.3, we choose Energy as the diversity measure. Formally, we optimize and report the following measure:

$$\frac{1}{N(N-1)} \sum_{i \neq j} \frac{1}{D(G_i, G_j) + \epsilon},$$

where $\epsilon$ is a small constant added for numerical stability. As soon as we fix the diversity measure that we rely on, the goal of each algorithm is to optimize this measure. In other words, in contrast to standard machine learning problems, we do not face the problem of overfitting.

We evaluate the following approaches described in Section 3: Greedy, Genetic, local optimization (LocalOpt), and iterative graph generative modeling (IGGM). Our evaluation also includes the comparisons against simple baseline models, specifically the Erdős-Rényi graphs sampled with various $p$ (ER-mix) and a sample from diverse random graph generators described in Section C.2. As an additional illustration, we also include a sample of graphs generated by the GraphWorld benchmark (Palowitch et al., 2022), where we vary the model parameters to increase diversity of the obtained graphs. In most of the experiments, we generate $N = 100$ graphs with $n = 16$ nodes. We also conduct experiments with non-neural algorithms on the set of 100 graphs with size $n = 64$.

Let us note that the algorithms introduced in Section 3 can be easily combined: the output of one algorithm can serve as an input to another. Thus, we evaluate the combinations of the algorithms. We use the notation '→' to denote the transition between the consecutive algorithms. Note that Greedy is the only strategy that does not generate any new graphs. Hence, its initial set should be already sufficiently diverse. Thus, we use graphs generated by diverse random graph models described in Section C.2.

We assume that for most algorithms, the most time-consuming operation is computing a graph representation (that is used for distance computations). Therefore, all algorithms except IGGM use the total limit of 3M generated graphs. For IGGM, the number of generated graphs is limited to 1M since training the graph generative model is time-consuming. In the tables, we use the square brackets to denote the number of computed graph representations for an algorithm or sub-algorithm.

**Numerical comparison**  In this section, we numerically analyze how well different approaches optimize the chosen diversity measure. Table 1 shows the results for selected algorithms and baselines. For more algorithms, please refer to Table 4 in Appendix, where we also report the standard deviation.

First, we note that all the proposed algorithms significantly improve the performance of the basic algorithms ER-mix and Random Graph Generators. Similarly, the diversity of GraphWorld is far from optimal. This is not surprising since GraphWorld does not directly optimize the diversity of graph structures and relies on the relatively simple stochastic block model.

Among the non-neural algorithms, the best performance is achieved by a combination of Greedy, Genetic, and LocalOpt (applied in this order). Such a combination is natural: Greedy starting from a set generated by different random graph generators is the simplest way to get an initial diverse set of graphs. Then, Genetic uses enough randomness to create all kinds of graph patterns to choose from. After that, LocalOpt is used to make final tuning with small graph modifications. In turn, the neural-network-based method IGGM gives a significant boost in diversity for GCD and Portrait-div distances and exhibits comparative results for NetLSD-heat. Note that it uses less budget for generated graphs but also requires training a graph generative neural model several times.

Table 1: Energy optimization results; see Table 4 in Appendix for the extended results

| Setup | GCD | Portrait-div | NetLSD-heat | NetLSD-wave |
|---|---|---|---|---|
| ER-mix | 0.281 | 43.057 | 72.387 | 0.583 |
| GraphWorld | 0.466 | 3.917 | 5.108 | 0.621 |
| Random Graph Generators | 0.553 | 6.009 | 116.685 | 1.334 |
| Greedy[3M] | 0.156 | 1.274 | 0.681 | 0.123 |
| ER-mix→Genetic[3M] | 0.139 | 1.264 | 0.677 | **0.117** |
| Greedy[1M]→Genetic[2M] | 0.139 | 1.263 | 0.674 | 0.118 |
| ER-mix→Genetic[1M]→LocalOpt[2M] | 0.138 | 1.259 | 0.675 | **0.117** |
| Greedy[1M]→LocalOpt[2M] | 0.139 | 1.255 | 0.679 | 0.118 |
| Greedy[1M]→Genetic[1M]→LocalOpt[1M] | 0.135 | 1.245 | **0.673** | **0.117** |
| IGGM[1M] | **0.120** | **1.213** | 0.675 | 0.148 |

Table 2: Diversity measured by Average; the graphs are the same as in Table 1

| Setup | GCD | Portrait-div | NetLSD-heat | NetLSD-wave |
|---|---|---|---|---|
| ER-mix | 4.350 | 0.607 | 0.936 | 6.302 |
| GraphWorld | 2.510 | 0.317 | 1.270 | 2.784 |
| Random Graph Generators | 2.059 | 0.212 | 0.025 | 1.190 |
| Greedy[3M] | 6.901 | 0.819 | 3.067 | 10.099 |
| ER-mix→Genetic[3M] | 7.553 | 0.830 | 3.056 | **10.625** |
| Greedy[1M]→Genetic[2M] | 7.614 | 0.826 | 3.072 | 10.549 |
| ER-mix→Genetic[1M]→LocalOpt[2M] | 7.734 | 0.831 | 3.056 | 10.621 |
| Greedy[1M]→LocalOpt[2M] | 7.494 | 0.830 | 3.051 | 10.485 |
| Greedy[1M] → Genetic[1M] → LocalOpt[1M] | 7.835 | 0.836 | **3.073** | 10.485 |
| IGGM[1M] | **8.687** | **0.854** | 3.066 | 10.364 |

Let us note that the basic algorithms in Table 1 (above the line) are not designed to optimize Energy and thus may accidentally generate pairs of graphs that are very close to each other, leading to significantly worse diversity. Hence, as an additional illustration of our results, we also report the average pairwise distance for the same sets of graphs. The results are shown in Table 2 and they are consistent with Table 1.

We also conducted additional experiments on larger graphs with $n = 64$ nodes. The results are shown in Table 5 in Appendix, and they are consistent with the results on smaller graphs.

**Examples of generated graphs** Since in our main experiments we generated 100 graphs, each having only 16 nodes, it is possible to visually inspect the generated graphs. To show that the generated graphs have very different structural patterns, we show some examples in Figure 1. This sample of graphs is chosen from the resulting set of the Genetic algorithm with diversity based on Portrait-div. It is clear that graphs vary in density, internal structure, number of cycles, and planarity. Importantly, these graphs are clearly distinct from the input distribution ER-mix. More examples showing all generated graphs are shown in Figures 7-11. We see that when combined with Portrait-div, both Genetic and IGGM generate visually diverse and interesting structures. One can also notice that NetLSD tends to generate many extremely sparse graphs, while GCD generates more dense graphs.

**Analysis of structural characteristics** Additionally, we analyze the structural characteristics of generated graphs. Figure 4 visualizes various characteristics for the ER-mix baseline, IGGM, and the combination of Greedy, Genetic, and LocalOpt. Obtaining a set of graphs in which an individual characteristic is diverse is easy: this can be achieved with the basic ER-mix. Hence, we visualize the joint distributions of pairs of characteristics.

It is clearly seen that compared to ER-mix, our algorithms lead to significantly more diverse pairs of characteristics. Also, it is worth mentioning that we often should not expect to cover all possible combinations: for instance, if the average degree is close to its maximal achievable value $n - 1$, then the clustering coefficient has to be close to 1. For more algorithms and combinations of characteristics, please refer to Figure 12 in Appendix.

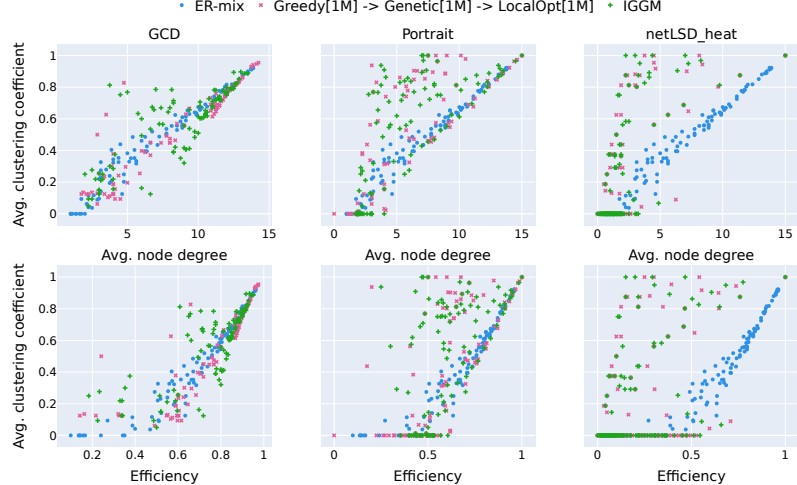

Figure 4: Joint distribution of graph characteristics for GCD, Portrait-div, NetLSD-heat

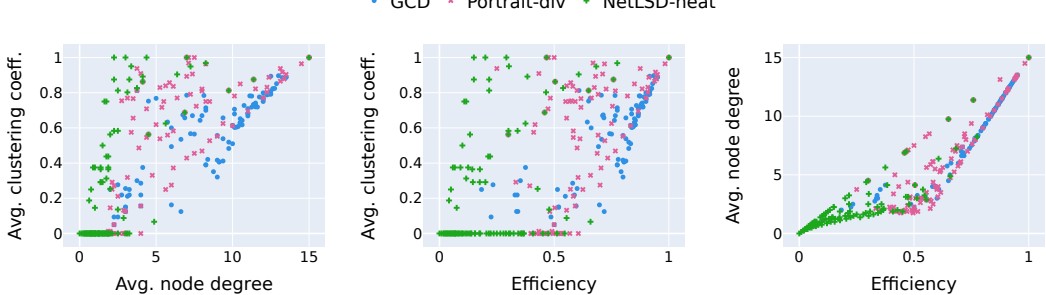

Figure 5: Joint distribution of graph characteristics for graphs from IGGM: comparing graph distances

**Comparing graph distances** Visualizing pairwise graph characteristics can help in the analysis and comparison of different graph distances. Indeed, the generated graphs significantly depend on a particular graph distance used for computing diversity. We visualize this in Figure 5. One observation that we make is that NetLSD is significantly biased towards sparse graphs (for clarity, Figure 5 shows the results for NetLSD-heat, but the wave variant has the same patterns). Indeed, for most of the generated graphs, the clustering coefficient is zero. Similarly, the average degree is usually small. However, despite this, the remaining NetLSD graphs may cover diverse combinations of characteristics. The fact that NetLSD is biased towards sparse graphs can also be seen in Figures 9-10, where we visualize the generated graphs. Then, Figure 5 shows the differences between GCD and Portrait-div. For instance, Portrait-div is significantly more diverse in terms of efficiency. This is natural, taking into account that GCD is based on local structures, while Portrait-div accounts for global characteristics.

## 5 Conclusion

In this work, we formulate the problem of generating structurally diverse graphs that can serve as representative instances for various graph-related tasks. We show that the problem is challenging as it is non-trivial to define what it means for a set of graphs to be diverse. In this regard, we propose desirable properties that a good diversity measure is expected to satisfy and choose a diversity measure based on them. Then, we show that random graph models do not provide sufficient diversity and propose various alternative approaches. Importantly, all the proposed algorithms can be applied to arbitrary diversity measures. Via a series of experiments, we show that the proposed approaches are capable of generating diverse graphs, both in terms of diversity measures and structural characteristics.

In this work, we have only made a first step to analyzing the problem of generating diverse graphs. There are plenty of promising directions for future research, and we hope that our work will encourage researchers to dive deeper into this problem. One particularly important challenge is scalability. If the number of nodes $n$ becomes large, then the number of possible graphs grows very fast, and for some methods (e.g., LocalOpt that uses single edge modifications) covering the whole space may become infeasible. Secondly, we believe that more advanced algorithms will be developed in the future. Also, further discussions on how to measure diversity and how to choose a proper graph distance seem to be very useful. Finally, it would be great to see practical applications of diverse graphs.

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

# A   Measuring diversity of a set of elements

## A.1   Related work on diversity measures

The concept of diversity is useful for various applications such as image and molecule generation or recommender systems. Diversity can be used to evaluate how representative is a given dataset, how diverse is a generated set, or for choosing a representative subset from a dataset (diversity sampling). In this section, we discuss relevant studies on measuring diversity.

A recent paper by Friedman and Dieng (2023) suggests measuring diversity via the Vendi Score (VS). This score requires a kernel function defined on pairs of elements. Then, a similarity matrix consisting of the pairwise kernel values is constructed and the Vendi Score is defined as the exponential of the Shannon entropy of the eigenvalues of the normalized similarity matrix.

Another recent work (Xie et al., 2023) investigates the problem of measuring the coverage (diversity) of a set $S$ in the context of molecular generation. The authors consider the following known measures: the average, maximum, and minimum pairwise distances. Xie et al. (2023) propose three axioms that a good measure of coverage is expected to satisfy: monotonicity, subadditivity, and dissimilarity. Then, they show that none of the abovementioned measures satisfy all three axioms simultaneously and propose a new coverage measure called #Circles. #Circles equals the maximum number of disjoint circles with centers placed at the elements of the set. However, while satisfying all the properties, this measure has two disadvantages. First, the complexity of calculating this measure is exponential. Second, it requires the radius parameter to be defined and a good value depends on a dataset. In our setup when the set of graphs dynamically changes during the optimization, this measure cannot be applied.

Let us discuss how the properties from Xie et al. (2023) relate to our study. *Monotonicity* and *subadditivity* describe the behavior of a measure when the size of the set increases and thus are not relevant in our setup. Hence, only the dissimilarity axiom remains. This axiom requires that the diversity of a pair of elements monotonically increases if one of them moves apart from the other. Our *monotonicity* property generalizes this axiom.

## A.2   Properties of popular measures

In this section, we revisit existing measures of diversity and show that typically used ones do not satisfy our requirements.

Table 3 lists diversity measures discussed in previous studies (Xie et al., 2023; Friedman and Dieng, 2023). Here we assume that $S = \{G_1, \ldots, G_N\}$. There are six well-known diversity measures defined over pairwise distances (Average, Diameter, and Bottleneck with two types of aggregation). Then, #Circles is the measure proposed in Xie et al. (2023) that also depends on the pairwise distances. Finally, Vendi Score is proposed in Friedman and Dieng (2023) and it is defined in terms of the pairwise kernels. We give the complete definition of Vendi Score below.

Table 3: Some known diversity measures

| Measure | Formula |
|---|---|
| Average | $\frac{2}{N(N-1)} \sum\limits_{i \neq j} D(G_i, G_j)$ |
| SumAverage | $\frac{1}{N} \sum\limits_{i \neq j} D(G_i, G_j)$ |
| Diameter | $\max\limits_{i \neq j} D(G_i, G_j)$ |
| SumDiameter | $\sum\limits_{i} \max\limits_{j \neq i} D(G_i, G_j)$ |
| Bottleneck | $\min\limits_{i \neq j} D(G_i, G_j)$ |
| SumBottleneck | $\sum\limits_{i} \min\limits_{j \neq i} D(G_i, G_j)$ |
| #Circles$(t), t \geq 0$ | $\max\limits_{C \subseteq [N]} |C|$ s.t. $D(G_i, G_j) > t\, \forall i \neq j \in C$ |
| Vendi Score | $\exp\left(-\sum\limits_{i=1}^{N} \lambda_i \log(\lambda_i)\right)$ |

**Theorem A.1.** *Among the previously used measures listed in Table 3, monotonicity is satisfied only by Average and SumAverage, while Uniqueness is satisfied only by Bottleneck.*

*Proof.* For each measure, we check whether it satisfies *monotonicity* and *uniqueness* defined in Section 2.

**Average**   $\frac{2}{N(N-1)} \sum\limits_{i \neq j} D(G_i, G_j)$

The monotonicity property is obviously satisfied.

We prove that uniqueness is not satisfied by the following example. Consider a multiset consisting of four different values: $\{0, 10, 11, 12\}$. The distances between the values are induced from the real line. The diversity of this set is $\frac{2}{4 \cdot 3}(10 + 11 + 12 + 1 + 2 + 1) = \frac{37}{6}$. If we replace 10 by the second copy of 0, we get a multiset $\{0, 0, 11, 12\}$ with a larger value of diversity $\frac{2}{4 \cdot 3}(0 + 11 + 12 + 11 + 12 + 1) = \frac{47}{6}$.

**SumAverage**   $\frac{1}{N} \sum_{i \neq j} D(G_i, G_j)$

Since SumAverage equals Average multiplied by $N$, their properties are similar: monotonicity is obviously satisfied for SumAverge and the same example gives the contradiction for uniqueness.

**Diameter**   $\max_{i \neq j} D(G_i, G_j)$

We prove that monotonicity is not satisfied by the following example. Consider one multiset with three elements and pairwise distances $10, 7, 4$ and another multiset with tree elements and pairwise distances $10, 7, 5$. Monotonicity requires the second set to have larger diversity, but the diversity of both sets is equal to 10.

To show that uniqueness is not satisfied, we consider the following example. Take a multiset consisting of three different elements: $\{0, 1, 2\}$. The distances between the elements are induced from the real line. The diversity of this set is 2. If we replace 1 with the second copy of 0, we get a multiset $\{0, 0, 2\}$ with the same diversity 2.

**SumDiameter**   $\sum_i \max_{j \neq i} D(G_i, G_j)$

We prove that monotonicity is not satisfied by the following example. Consider one multiset with three elements and pairwise distances $10, 7, 4$ and another multiset with tree elements and pairwise distances $10, 7, 5$. Monotonicity requires the second set to have larger diversity, but the diversity of both sets is equal to $10 + 10 + 7 = 27$.

We prove that uniqueness is not satisfied by the following example. Consider a multiset consisting of three different elements: $\{0, 1, 2\}$. The distances between the elements are induced from the real line. The diversity of this set is $2 + 2 + 1 = 5$. If we replace 1 with the second copy of 0, we get a multiset $\{0, 0, 2\}$ with larger diversity $2 + 2 + 2 = 6$.

**Bottleneck**   $\min_{i \neq j} D(G_i, G_j)$

To show that monotonicity is violated, consider a multiset consisting of three different elements: $\{0, 1, 5\}$. The distances between the elements are induced from the real line. The diversity of this set is 1. Monotonicity requires the set $\{0, 1, 6\}$ to have larger diversity. But the diversity of the set $\{0, 1, 6\}$ is also equal to 1.

Uniqueness is satisfied. Indeed, any multiset with pairwise different elements has diversity greater than 0, and any multiset with two copies of one element has diversity 0.

**SumBottleneck**   $\sum_i \min_{j \neq i} D(G_i, G_j)$

To show that monotonicity is violated, consider a multiset consisting of four different elements: $\{0, 1, 5, 6\}$. The distances between the elements are induced from the real line. The diversity of this set is $1 + 1 + 1 + 1 = 4$. Monotonicity requires the set $\{0, 1, 7, 8\}$ to have larger diversity. But the diversity of the set $\{0, 1, 7, 8\}$ is also equal to $1 + 1 + 1 + 1 = 4$.

We prove that uniqueness is not satisfied by the following example. Consider a multiset consisting of four different elements: $\{0, 1, 9, 10\}$. The distances between the elements are induced from the real line. The diversity of this set is $1 + 1 + 1 + 1 = 4$. If we replace 1 with the second copy of 9, we get a multiset $\{0, 9, 9, 10\}$ with larger diversity $9 + 0 + 0 + 1 = 10$.

**#Circles**$(t)$   $\max_{C \subseteq [N]} |C|$ s.t. $D(G_i, G_j) > t \, \forall i \neq j \in C$

We prove that monotonicity is not satisfied by the following example. Consider one multiset with three elements and pairwise distances $10, 7, 4$ and another multiset with tree elements and pairwise distances $10, 8, 4$. Monotonicity requires the second set to have larger diversity, but: for $t < 4$ the diversity of both sets is equal to 3; for $4 \leq t < 10$ the diversity of both sets is equal to 2; for $10 \leq t$ the diversity of both sets is equal to 1.

Now, we fix $t$ and prove that uniqueness is not satisfied by the following example. Consider a multiset consisting of three different elements: $\{0, \frac{t}{2}, t\}$. The distances between the elements are induced from the real line. The diversity of this set is 2. If we replace $\frac{t}{2}$ by the second copy of 0, we get a multiset $\{0, 0, t\}$ with the same diversity 2.

**Vendi Score** $\quad \exp\left(-\sum_{i=1}^{N} \lambda_i \log(\lambda_i)\right)$

First, let us give a formal definition of this measure.

**Definition A.2** (Vendi Score, Friedman and Dieng (2023)). Let $S = \{G_1, \ldots, G_N\}$ be a multiset and let $k : S \times S \to \mathbb{R}$ be a similarity function, such that $\forall i : k(G_i, G_i) = 1$ and the matrix $K \in \mathbb{R}^{N \times N}$ defined by $K_{i,j} := k(G_i, G_j)$ is positive-semidefinite and symmetric. Denote by $\lambda_1, \ldots \lambda_N$ the eigenvalues of the matrix $K/N$. Then, the Vendi Score is defined as the exponential of the Shannon entropy of the eigenvalues of $K/N$:

$$\exp\left(-\sum_{i=1}^{N} \lambda_i \log(\lambda_i)\right), \tag{6}$$

where we use the convention $0 \log 0 = 0$.

Our monotonicity property uses distances instead of similarities, but we can naturally reformulate it in terms of pairwise similarities by replacing the condition $D(G_i, G_j) \leq D(g(G_i), g(G_j))$ with the condition $k(G_i, G_j) \geq k(g(G_i), g(G_j))$. Our uniqueness does not use the notion of distance, so we can use it as is. Thus, we can check whether the Vendi Score satisfies monotonicity and uniqueness.

We prove that monotonicity is not satisfied with the following example. Consider two positive-semidefinite symmetric matrices:

$$K_1 = \begin{pmatrix} 1 & 0.1 & 0.8 \\ 0.1 & 1 & 0.4 \\ 0.8 & 0.4 & 1 \end{pmatrix}, \quad K_2 = \begin{pmatrix} 1 & 0.2 & 0.8 \\ 0.2 & 1 & 0.4 \\ 0.8 & 0.4 & 1 \end{pmatrix}. \tag{7}$$

Monotonicity requires $K_1$ to have higher diversity than $K_2$. But Vendi Score of $K_1$ is 2.203 and Vendi Score of $K_2$ is $2.212 > 2.203$.

To show that uniqueness is not satisfied, consider the following example. Take two positive-semidefinite symmetric matrices:

$$K_1 = \begin{pmatrix} 1 & 0.6 & 0.2 \\ 0.6 & 1 & 0.9 \\ 0.2 & 0.9 & 1 \end{pmatrix}, \quad K_2 = \begin{pmatrix} 1 & 1 & 0.2 \\ 1 & 1 & 0.2 \\ 0.2 & 0.2 & 1 \end{pmatrix} \tag{8}$$

If Vendi Score has uniqueness property, then $K_1$ must have higher diversity than $K_2$. But the Vendi Score of $K_1$ is 1.81 and Vendi Score of $K_2$ is $1.86 > 1.81$.

$\square$

## A.3 Illustrating the effect of the Energy parameter $\gamma$

To illustrate the effect of $\gamma$, let us consider a simple setup when points are distributed in a square. We place 50 points uniformly at random in a unit square and optimize Energy for $\gamma = 0.1, 0.3, 0.5, 1, 2, 3, 10$. The results are shown in Figure 6. Clearly, for small $\gamma$ the coverage of the non-boundary region is not sufficient. However, for larger values (including $\gamma = 1$) the distribution looks sufficiently diverse.

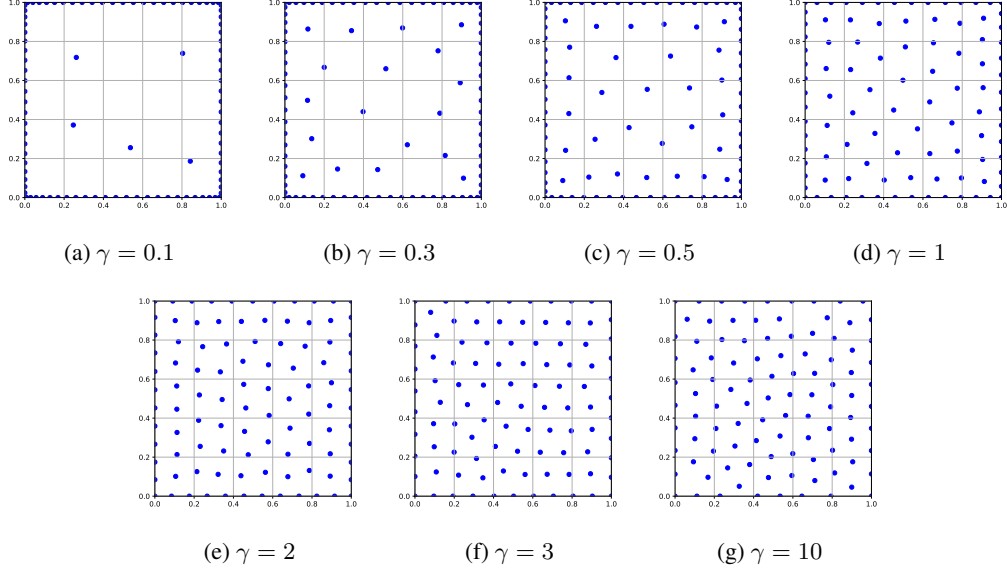

$$\text{(a) } \gamma = 0.1 \qquad \text{(b) } \gamma = 0.3 \qquad \text{(c) } \gamma = 0.5 \qquad \text{(d) } \gamma = 1$$

$$\text{(e) } \gamma = 2 \qquad \text{(f) } \gamma = 3 \qquad \text{(g) } \gamma = 10$$

Figure 6: The effect of the Energy parameter $\gamma$ for points distributed in a square

### A.4 Unboundedness of Energy

When two elements of a set get closer to each other, Energy can become arbitrarily large. Such behavior can cause some interpretability problems when we compare the obtained Energy for different algorithms. In terms of the desirable properties, the issue is that both monotonicity and uniqueness can be violated if not all elements are different. However, it turns out that if one requires monotonicity and uniqueness to be satisfied even when some elements coincide, then it becomes very challenging to construct a measure satisfying all the desirable properties. This problem was recently addressed by Mironov and Prokhorenkova (2024) who require monotonicity and uniqueness to be satisfied for all the initial configurations and also add an important property of continuity. The authors construct two examples of measures satisfying all the properties, but both of them are NP-hard to compute and thus are infeasible to use in practice. Whether there exists a computationally feasible measure satisfying all the properties is currently unknown.

Thus, we have chosen Energy as the best option available. The advantage of Energy is that when it is optimized, the obtained distribution is indeed diverse (see, e.g., Figure 6). In other words, Energy can be degenerate for configurations that are not sufficiently diverse, but it can be used to compare algorithms that optimize diversity. Also, recall that for numerical stability, we add a small constant to the denominator of Energy, so it does not go to infinity in practice.

Importantly, for the completeness of our study, in Table 2 we report the average pairwise distance. As discussed above, this measure cannot be used as a function that is optimized by an algorithm since it can lead to degenerate solutions. That is why we only use it as an assistive measure and apply it to the sets of graphs obtained by optimizing other objectives.

## B   Considered graph distances

In this section, we describe the graph distances that we use in our experiments.

**NetLSD (Tsitsulin et al., 2018)**   NetLSD treats a graph as a dynamic system and simulates heat and wave diffusion processes on nodes and edges of a given graph, followed by measuring system conditions at fixed timestamps. More formally, let $\lambda_j$ be the $j$-th smallest eigenvalue of the normalized Laplacian of a graph $G$. For a timestamp $t$, we define *heat trace* $h_t$ and *wave trace* $w_t$ of a graph $G$ as follows:

$$h_t = \sum_j e^{-t\lambda_j}, \ \ w_t = \sum_j e^{-it\lambda_j} . \tag{9}$$

Here $t > 0$ for the heat trace and $t \in [0, 2\pi)$ for the wave trace.

Then, the *heat trace signature* and *wave trace signature* of $G$ are defined as the collections of the corresponding traces at different timestamps, i.e., $h(G) = \{h_t\}_{t \in \mathcal{T}_h}$ and $w(G) = \{w_t\}_{t \in \mathcal{T}_w}$. As in the original article, we use 250 log-spaced timestamps between $10^{-2}$ and $10^2$ for $\mathcal{T}_h$ and 250 equally-spaced timestamps between 0 and $2\pi$ for $\mathcal{T}_w$, respectively.

Finally, the NetLSD distance (heat or wave) between two graphs $G$ and $G'$ is computed as any distance measure between the corresponding signatures. Following Tsitsulin et al. (2018), we use the Euclidean distance.

**Graphlet Correlation Distance (GCD) (Yaveroğlu et al., 2014)** Graphlet Correlation Distance computes the distance between two graphs based on their graphlet statistics. Here graphlets are defined as connected graphs with 2, 3, or 4 nodes, with one of them marked. There are exactly 15 such graphs.

Consider any graph $G$ with $n$ nodes. Choose any node $v \in G$ and any graphlet $R$. We count the number of subgraphs in $G$ which are isomorphic to $R$, such that the marked node of $R$ coincides with $v$. Doing this for fixed $v$ and all graphlets, we get 15 numbers corresponding to $v$. These numbers are not independent: if we know the counts for some graphlets, we can find the counts for some other graphlets. Getting rid of 4 redundant counts, we are left with 11 values (and the corresponding graphlets). So now we have a vector of length 11 for each node of $G$. We combine these vectors in a matrix $L$ with $n$ rows and 11 columns. Using this matrix $L$, we compute $11 \times 11$ *Graphlet Correlation Matrix* (GCM) as follows. The cell $(i, j)$ of GCM contains Spearman's correlation coefficient between $i$-th and $j$-th columns of the matrix $L$. In other words, this cell contains the correlation between the number of times a node is a part of the $i$-th graphlet and the number of times the same node is a part of the $j$-th graphlet.

The *graphlet correlation distance* between two graphs $G$ and $G'$ is then computed as the Euclidean distance between the upper-triangular parts of their GCMs:

$$D(G, G') = \sqrt{\sum_{1 \leq i < j \leq 11} \left( GCM_G(i, j) - GCM_{G'}(i, j) \right)^2}. \tag{10}$$

**Portrait Divergence (Bagrow and Bollt, 2019)** The *network portrait* (Bagrow et al., 2008) of a graph $G$ is a matrix $B$ with elements $b_{lk}$ being the number of such nodes $v$ that there are exactly $k$ nodes at a distance $l$ from $v$. This matrix captures both local and global graph statistics. Based on the portrait $B$, one can compute the joint probability of choosing a pair of nodes at a distance $l$ from each other and that the first node has $k$ nodes at a distance $l$ from it:

$$P_B(k, l) = P(k|l)P(l) = \left( \frac{\sum_{k'=0}^{n} k' b_{lk'}}{n} \right) \frac{b_{lk}}{\sum_c n_c^2}, \tag{11}$$

where $\sum_c n_c^2$ is the normalization over the sizes $n_c$ of the connected components. Then, the *portrait divergence* between two graphs $G$ and $G'$ is computed as the Jensen-Shannon divergence of the distributions $P_{B(G)}(k, l)$ and $P_{B(G')}(k, l)$.

## C  Algorithms for diversity optimization

In this section, we give more details on the algorithms that we use for diversity optimization. Recall that our algorithms assume that the overall diversity of a set of graphs $\{G\} \cup S$ can be written as $\text{Diversity}(\{G\} \cup S) = g(f(G, S), c(S))$, where $g$ is a function that is monotone w.r.t. its arguments. The function $f(G, S)$ is called *fitness* of a graph $G$ w.r.t. a set of graphs $S$.

The code of our experiments is publicly available at `https://github.com/Abusagit/Challenges-on-generating-structurally-diverse-graphs`.

### C.1  Greedy algorithm

**Algorithm** Assume that we are given a pre-generated set $\hat{S}$ of $M$ graphs with diverse structural properties, $M \gg N$. Let $S$ be our constructed set of diverse graphs which is initially an empty set.

Then, our greedy algorithm consists of $N$ steps. We start with $S = \emptyset$ and at the first step we select a graph from the set $\hat{S}$ uniformly at random to be the first element in $S$. At each subsequent step, we choose $G \in \hat{S}$ with the maximal value of the fitness $f(G, S)$. Then, we add this graph $G$ to the set $S$. After $N$ steps, we get a set $S$ of size $N$, which is our approximation of the maximally diverse set.

**Analysis** While being very simple, the greedy algorithm turns out to be very effective when supplied with a sufficiently diverse set $\hat{S}$. The following theorem provides the bounds on diversity. While in this paper we focus on the Energy diversity measure, we also provide the results for Average and Bottleneck measures.

**Theorem C.1.** *Assume that the diversity function is Energy, Average, or Bottleneck. Let $\hat{S}$ be a set of graphs. If the greedy algorithm selected a subset $S$ from $\hat{S}$, then*

$$\text{Diversity}(S) \geq \frac{1}{2}\text{Diversity}(\bar{S}) \text{ for Average and Minimum,}$$

$$\text{Diversity}(S) \geq 2^\gamma \text{Diversity}(\bar{S}) \text{ for Energy}(\gamma),$$

*where $\bar{S}$ is the maximally diverse subset of $\hat{S}$ satisfying $|\bar{S}| = |S| = N$.*

*Proof.* Let us prove the statement for each of the diversity measures.

**Average** Suppose $\max\limits_{G_1, G_2 \in \hat{S}} D(G_1, G_2) = d$ and this maximum is achieved for $G_1 = V_1, G_2 = V_2$ for some $V_1, V_2 \in \hat{S}$.

Since all pairwise distances between the elements of $\bar{S} \subset \hat{S}$ are less than or equal to $d$, the average pairwise distance between the elements of $\bar{S}$ is also less than or equal to $d$. That is, $\text{Diversity}(\bar{S}) \leq d$. Therefore, to prove the theorem it is sufficient to prove that for the greedily picked $S$ we have $\text{Diversity}(S) \geq \frac{d}{2}$. We prove it by induction on $N$, which is the size of the set $S$. The base case $N = 2$ is trivial, since for every element there exists a second element at distance more than or equal to $\frac{d}{2}$ (by the triangle inequality, one of the ends of any diameter can serve as such a element).

Inductive step. Suppose the induction statement holds for $N = k$. Let us prove it for $N = k + 1$. Suppose the greedy algorithm picked graphs $A_1, \ldots, A_k, A_{k+1}$ in this order. We need to prove that the average pairwise distance between $A_1, \ldots, A_k, A_{k+1}$ is at least $\frac{d}{2}$. By induction hypothesis, the average pairwise distance between $A_1, \ldots, A_k$ is at least $\frac{d}{2}$. So, it is sufficient to prove that the average distance between $A_{k+1}$ and $A_1, \ldots, A_k$ is at least $\frac{d}{2}$. This is equivalent to $\sum\limits_{i=1}^{k} D(A_i, A_{k+1}) \geq \frac{kd}{2}$.

By the triangle inequality, we have $D(A_i, V_1) + D(A_i, V_2) \geq D(V_1, V_2) = d$ for all $1 \leq i \leq k$. Summing these inequalities for all $i$, we get

$$\sum_{i=1}^{k} D(A_i, V_1) + \sum_{i=1}^{k} D(A_i, V_2) \geq kd,$$

therefore $\sum\limits_{i=1}^{k} D(A_i, V_1) \geq \frac{kd}{2}$ or $\sum\limits_{i=1}^{k} D(A_i, V_2) \geq \frac{kd}{2}$. Thus, by the construction of the greedy algorithm, we have $\sum\limits_{i=1}^{k} D(A_i, A_{k+1}) \geq \frac{kd}{2}$.

**Bottleneck** Suppose $\bar{S} = C_1, \ldots, C_N$ and $\text{Diversity}(\bar{S}) = m$. Suppose the greedy algorithm has already made $k < N$ steps choosing graphs $A_1, \ldots, A_k$, and the diversity of $A_1, \ldots, A_k$ is at least $\frac{m}{2}$. We define an *open ball* with center in $G \in \hat{S}$ and radius $r$ as the set of all graphs in $\hat{S}$ such that their distance to $G$ is less than $r$. Consider $N$ open balls with centers in $C_1, \ldots, C_N$ and radius $\frac{m}{2}$. Clearly, these balls do not intersect. Since $k < N$, there is at least one of $N$ balls that does not contain any of $A_1, \ldots, A_k$. Therefore, the distance from the center of this ball to all $A_1, \ldots, A_k$ is at least $\frac{m}{2}$.

Thus, the greedy algorithm can make one more step while still preserving the diversity of the chosen set not less than $\frac{m}{2}$. Since we prove it for all $k < N$, the diversity of the greedy algorithm result on step $N$ will be at least $\frac{m}{2} = \frac{1}{2}\text{Diversity}(\bar{S})$.

**Energy** We prove that $\text{Diversity}(S) \geq 2^\gamma \text{Diversity}(\bar{S})$ by induction on $N$, which is the size of the set $S$. The base case $N = 2$ is trivial, since for every element there exists a second element at distance more than or equal to $\frac{d}{2}$ (by the triangle inequality, one of the ends of any diameter can serve as such a element).

Inductive step. Suppose the statement holds for $N = k$. Let us prove it for $N = k + 1$. Suppose the greedy algorithm made $k$ steps and picked graphs $A_1, \ldots, A_k$ in this order. Suppose $\bar{S} = C_1, \ldots, C_{k+1}$. We pair $A_1$ with the nearest graph from the set $\bar{S}$ (ties break randomly), w.l.o.g. we assume that this graph is $C_1$. We pair $A_2$ with the nearest graph from the set $\bar{S} \setminus \{C_1\}$, w.l.o.g. we assume that this graph is $C_2$. We pair $C_3$ with the nearest graph from the set $\bar{S} \setminus \{C_1, C_2\}$, w.l.o.g. we assume that this graph is $C_3$, etc. Note that the graph $C_{k+1}$ is left unpaired. Let us prove that $\text{Diversity}(\{A_1, \ldots, A_k, C_{k+1}\}) \geq 2^\gamma \text{Diversity}(\bar{S})$, from which the statement of the theorem follows trivially.

By the induction hypothesis, we have $\text{Diversity}(\{A_1, \ldots, A_k\}) \geq 2^\gamma \text{Diversity}(\{C_1, \ldots, C_k\})$. So, it is sufficient to prove that $-\sum_{i=1}^{k} \frac{1}{D(A_i, C_{k+1})^\gamma} \leq -2^\gamma \sum_{i=1}^{k} \frac{1}{D(C_i, C_{k+1})}$ which is equivalent to $\sum_{i=1}^{k} \frac{1}{D(A_i, C_{k+1})^\gamma} \geq 2^\gamma \sum_{i=1}^{k} \frac{1}{D(C_i, C_{k+1})^\gamma}$. Given that $\gamma > 0$, this inequality holds if we prove that $\frac{1}{D(A_i, C_{k+1})} \geq 2 \frac{1}{D(C_i, C_{k+1})}$ for every $i$. Rewriting the last inequality, we get $D(C_i, C_{k+1}) \leq 2D(A_i, C_{k+1})$. This inequality holds since:

$$D(C_i, C_{k+1}) \leq D(C_i, A_i) + D(A_i, C_{k+1}) \leq 2D(A_i, C_{k+1}).$$

Here the first inequality is the triangle inequality. The second inequality follows from the pairing construction which ensures that $D(C_i, A_i)$ is less than or equal to $D(A_i, C_{k+1})$. □

Now, let us analyze the time complexity of the greedy algorithm. Here and below, we assume that computing a graph representation for a distance $D$ for a graph with $n$ nodes requires $a$ numerical operations. Given that, the distance between two representations requires $b$ numerical operations. Every distance is computed only once and then the result is cached.

**Proposition C.2.** *The time complexity of the greedy algorithm is $O((a + bN)M)$, where $M = |\hat{S}|$ (the size of the initial population) and $N = |S|$ (the size of the desired diverse population).*

*Proof.* We do not take into account the time complexity of generating the population $\hat{S}$ since it heavily depends on the choice of graph generators. First, we compute the descriptors for all graphs in $\hat{S}$, which is $aM$ operations. For all graphs, we set their current fitness to 0. Then, at each step of the greedy algorithm, we compute the distances from all elements of $\hat{S} \setminus S$ to an element added to $S$ on this step, which is $O(bM)$ operations. Using the computed distances, we update the current fitness for all graphs in $O(M)$ operations. The choice of an element that we add to $S$ can be done in $O(M)$ operations. So, each step requires $O(bM)$ operations. Given that we have $N$ steps, the resulting time complexity of the algorithm is $O(aM + bMN) = O((a + bN)M)$. □

**Generating the set $\hat{S}$** Generating a sufficiently diverse set $\hat{S}$ is a necessary ingredient of the success of the greedy algorithm. To generate such a set, we use several random graph models with different properties. For each model, we iterate over the parameter combinations to get structurally different graphs. After this procedure, we assume that the set $\hat{S}$ is rich enough to contain a wide variety of graphs. The next subsection describes the models used for generating the set $\hat{S}$.

## C.2 Mixture of random graph generators

All models below generate graphs with a fixed number of nodes $n$. In our experiments, we take $n = 16$ and $n = 64$. The graphs are undirected and without self-loops and multiple edges. We describe only the versions of models that we use in our experiments. Some of these models have more general versions with more parameters that we do not use and do not describe. To obtain a sample of graphs, we generate an (approximately) equal number of graphs for each combination of a model and its parameters.

**Erdős-Rényi** In this model, each edge is included in the generated graph with probability $p$, independently from all other edges.

We consider $p \in \left\{ \frac{1}{16}, \frac{1}{8}, \frac{1}{4}, \frac{1}{2}, \frac{3}{4}, \frac{7}{8}, \frac{15}{16} \right\}$.

**Preferential Attachment (Barabási and Albert, 1999)** In preferential attachment models, nodes are added one by one, and each new node attaches to several previous ones with probabilities depending on their degrees. Here the parameter $m$ reflects the number of outgoing edges added together with each new node. The probability that a new node is attached to an older node $i$ is proportional to $k_i + \alpha$, where $k_i$ is the number of incoming edges of $i$ and $\alpha > 0$ is a parameter reflecting the attractiveness of nodes with zero incoming degrees.

We consider $m \in \{1, 2, 4\}$ and $\alpha \in \{m/2, m, 2m\}$. Such values of $\alpha$ give the power-law degree distribution with parameters $\gamma \in \{2.5, 3, 4\}$ (Ostroumova et al., 2013).

**Holme–Kim (Holme and Kim, 2002)** This is a modification of the preferential attachment model, allowing for varying the number of triangles. Again, nodes are added one by one; each node appears with $m$ edges. Edges are also added one by one, and there are two types of edges: random and triangle-forming. Random edges connect the new node with an old one with probability proportional to the total degree of the old node. Each random edge can be followed by a triangle-forming edge (with probability $p$) or by another random edge (otherwise). To add a triangle-forming edge, we do the following: we uniformly sample a neighbor of the previously chosen node and connect the new node to this neighbor.

In our experiments, we take $m \in \{2, 4\}$ and $p \in \{0.5, 1\}$.

**Random graph with power-law expected degree sequence (Chung and Lu, 2002)** First, we sample a sequence $W = (w_1, \ldots, w_n)$ from a power-law distribution with parameter $\gamma$. Then, we construct a graph by connecting the nodes $i$ and $j$ with probability $\frac{w_i w_j}{\sum_k w_k}$.

Here we use $\gamma \in \{2, 2.5, 3, 4\}$.

**Random geometric graph (Penrose, 2003)** First, $n$ nodes are placed uniformly at random in the unit cube in $\mathbb{R}^d$. Then, two nodes are joined by an edge if the distance between them is at most $r$.

We take $d \in \{2, 3\}$, where for $d = 2$ we have $r \in \{0.2, 0.3, 0.5\}$ and for $d = 3$ we have $r \in \{1/3, 0.5, 0.65\}$.

**Random regular graph** This model generates a random graph, each node of which has degree $d$.

We consider $d \in \{1, 2, 4, 8, 10\}$.

**Stochastic block model (Holland et al., 1983)** We divide $n$ nodes into $r$ sets (*blocks*) of (approximately) equal size. Each edge between two nodes from the same block is included with probability $p$, independently from all other edges. Each edge between two nodes from different blocks is formed with probability $q$, independently from all other edges.

In the experiments, we use $r = 2$ and $r = 3$. For $r = 2$, we use all combinations of pairs $(p, q)$ from the set $\{(2s, s) \mid \forall s \in S\} \cup \{(s, 2s) \mid \forall s \in S\}$, where $S \in \left\{ \frac{1}{16}, \frac{1}{8}, \frac{1}{4}, \frac{1}{2} \right\}$. For $r = 3$ we use $(p, q) \in \left\{ \left( \frac{1}{2}, \frac{1}{4} \right), \left( \frac{1}{5}, \frac{2}{5} \right) \right\}$.

Our implementations are based on *NetworkX* and *igraph* Python libraries.

## C.3 Genetic algorithm

This section describes our implementation of the genetic approach.

**High-level description of the algorithm** First, we obtain an initial population of $N$ graphs using either a specific random generator, or an ensemble of models, or user input with the size $N$. After that, we repeatedly do the following procedure. We choose two distinct parental graphs $P_1$ and $P_2$ from the population, and via crossover and mutation generate a child graph $C'$. After that, we go over all graphs in the current population, try to replace each of them with $C'$ and compute the difference

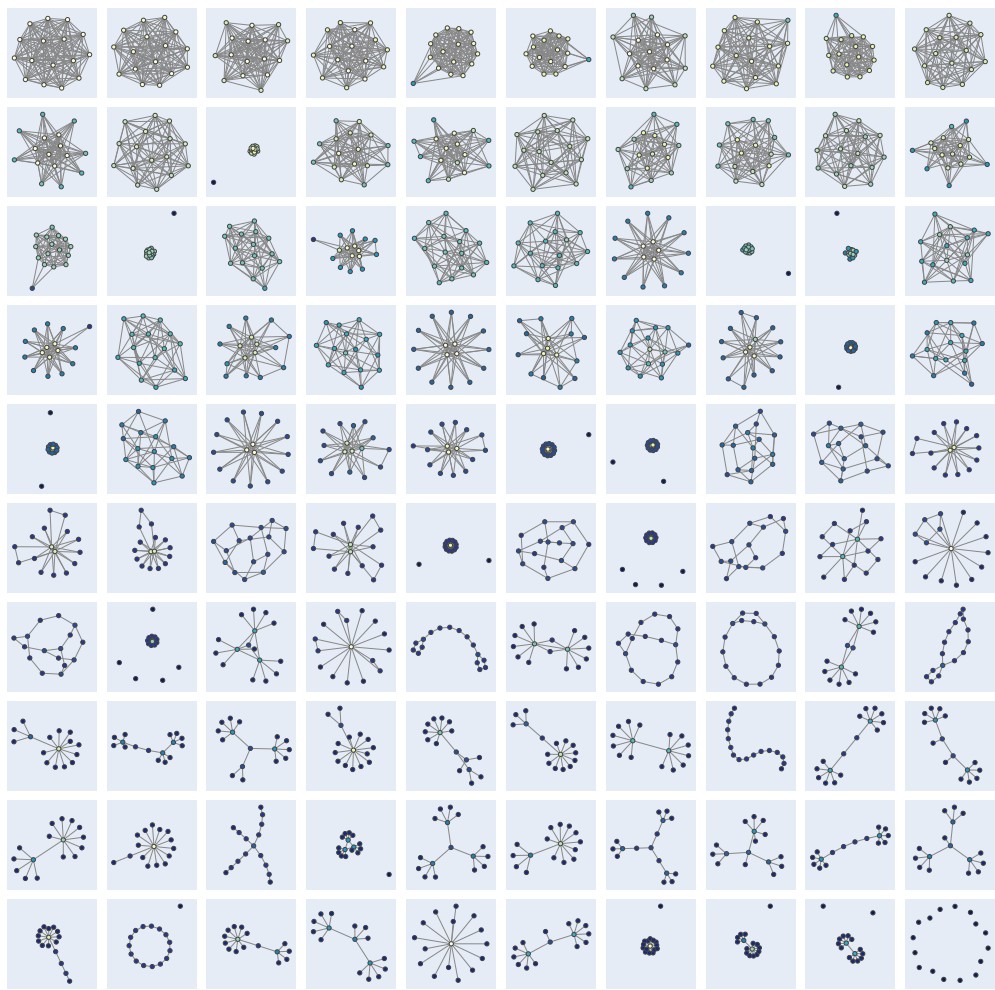

Figure 7: Graphs from Genetic with Portrait-div

in diversity caused by the replacement. Then, we choose the graph with the largest difference and if it is positive we replace it with $C'$; otherwise, we do nothing. Then we repeat the procedure. We also limit the number of unsuccessful attempts and denote this parameter as $K$. Now let us consider all stages of the algorithm in detail.

**Initial population** For an initial set of graphs, we can either use established random graph models (e.g., generate $N$ graphs with $n$ nodes each using the Erdős-Rényi model with edge probability $p = 0.5$ or with mixed $p$) or can pass an already saved set of graphs through command-line argument (for instance, the resulting population from the greedy algorithm can become the initial population for the genetic algorithm). The nodes of each graph are labeled by numbers $1, 2, \ldots, n$ (we will need these labels for the crossover procedure). Denote the generated population by $S$.

**Selecting parents** We randomly select two different graphs from $S$ as parents. To do so, we assign each graph the probability proportional to its fitness w.r.t. set of all other graphs and sample two different graphs from the resulting distribution.[4] We denote these graphs by $P_1$ and $P_2$.

---

[4]Since Energy is negative, for this measure we sample with probabilities proportional to $-\frac{1}{\text{fitness}}$.

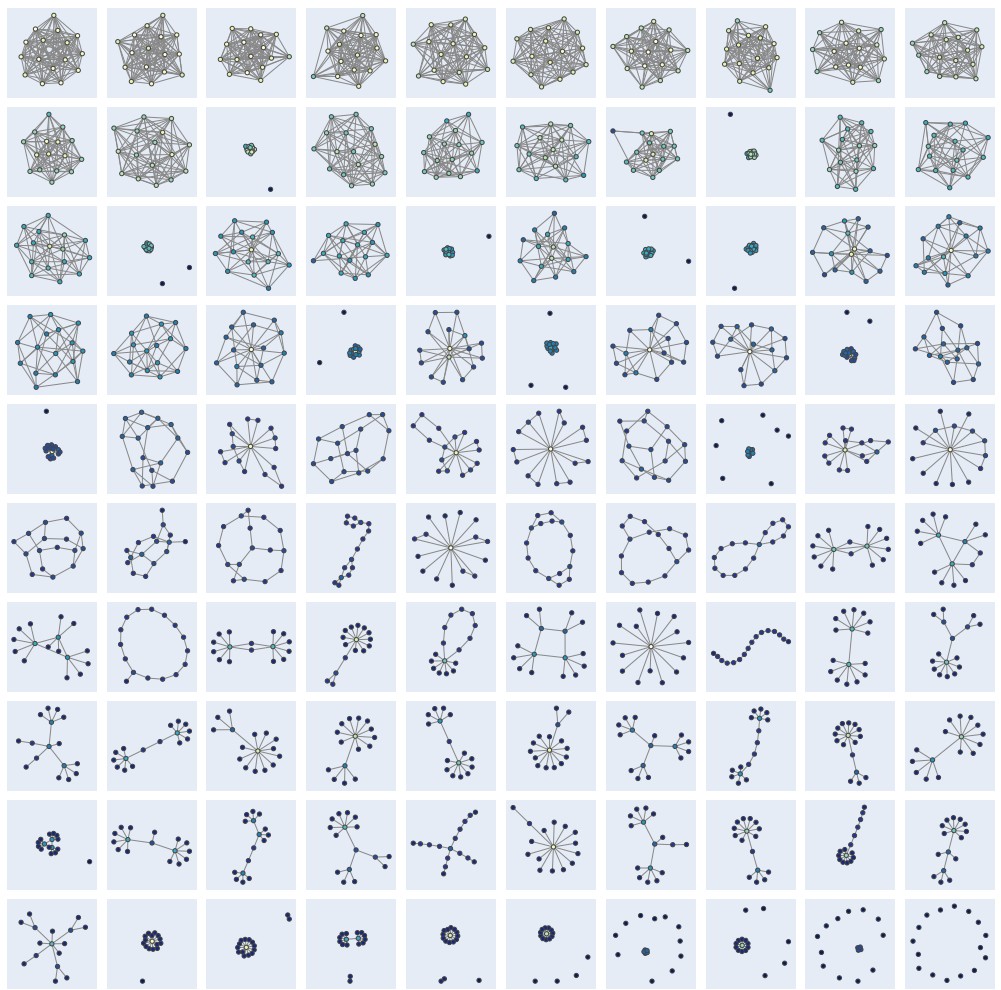

Figure 8: Graphs from IGGM with Portrait-div

**Crossover** Given the graphs $P_1$ and $P_2$, we construct a new child graph $C$. Informally, we want $C$ to copy some nodes and edges from $P_1$ and some nodes and edges from $P_2$. For each label from $1$ to $n$, we randomly and independently assign a number $1$ or $2$ with equal probability. If a label $i$ has been assigned $1$, then the node $i$ is copied from $P_1$, and otherwise it is copied from $P_2$. After that, for each pair of nodes $i < j$, we determine whether $C$ has an edge between $i$ and $j$ as follows:

- if $i$ was assigned $1$ and $j$ was assigned $1$, then $C$ has an edge between $i$ and $j$ iff $P_1$ has an edge between $i$ and $j$;

- if $i$ was assigned $2$ and $j$ was assigned $2$, then $C$ has an edge between $i$ and $j$ iff $P_2$ has an edge between $i$ and $j$;

- if $i$ and $j$ were assigned different numbers, then we randomly choose one parent, and $C$ has an edge between $i$ and $j$ iff the selected parent has an edge between $i$ and $j$.

**Mutation** We follow Ipsen and Mikhailov (2002) and perform mutations as follows. Given a graph $C$, with probability $\alpha$ we construct a mutated graph $C'$. First, we choose a node from $C$ uniformly at random and delete all edges connected to it. Then, we select a number $k \in \{1, 2, \ldots, n-1\}$ uniformly at random. We draw $k$ new edges from the node. These edges connect to random distinct nodes of the graph, with each node having an equal probability of being connected to. The resulting

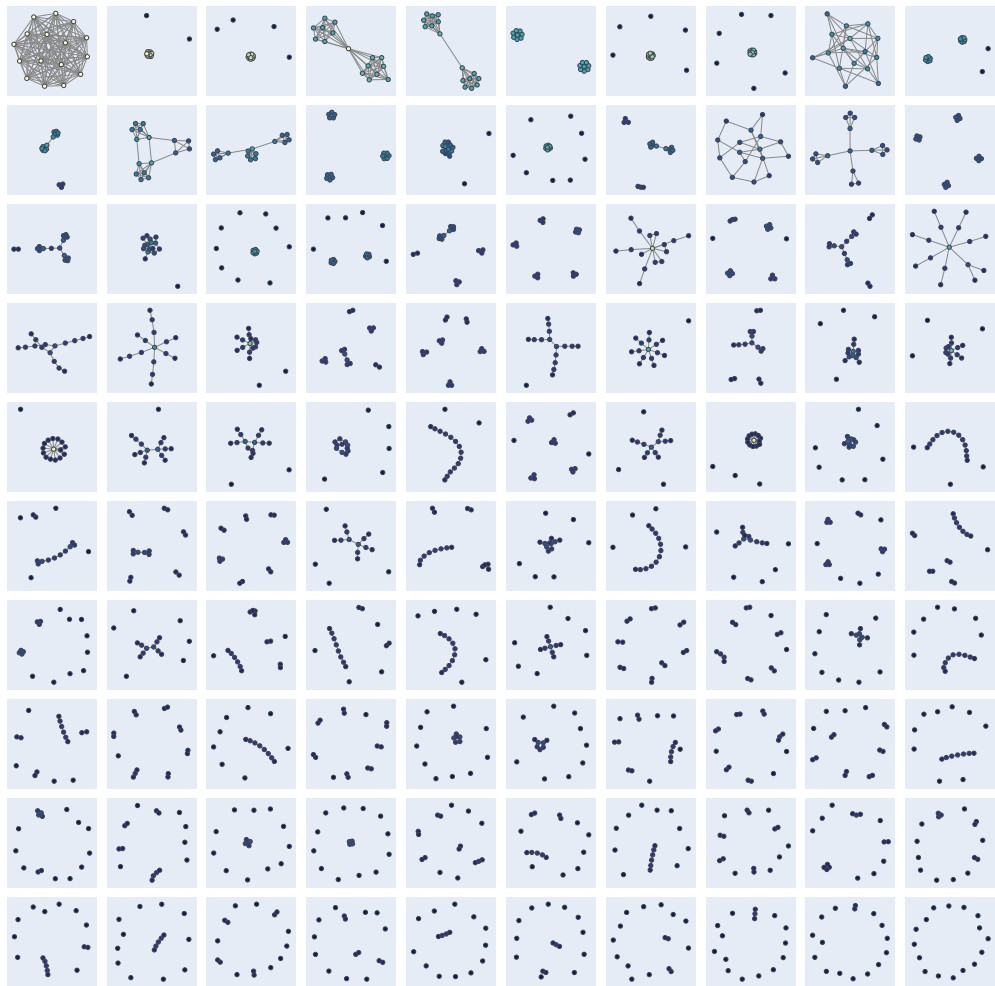

Figure 9: Graphs from IGGM with netLSD-heat: most of the graphs are sparse

graph is denoted by $C'$. The probability of mutation $\alpha$ is another parameter of the algorithm. If mutation does not occur, then we take $C' = C$.

**Update**  We check each graph from the population and try to replace it with $C'$. By $U$ we denote the graph giving the largest improvement after the replacement. If $f(U, S \setminus U) < f(C', S \setminus U)$, then we remove $U$ from $S$, add $C'$ to $S$, and call $C'$ a successful child. Otherwise, $C'$ is called unsuccessful, and the population $S$ remains unchanged. Note that this procedure does not decrease the diversity of the population and keeps the size of the population equal to $N$.

**Number of update attempts**  We limit the complexity of our algorithm by the total number of update attempts $L$, after which the algorithm finishes. Both successful and unsuccessful attempts are counted. To prevent the algorithm from getting stuck in a local optima, we count the number of consecutive failed updates. If there are no successful updates during the last $K$ attempts, we accept the candidate $C'_k$ among the last $K$ with the maximum value of $f(C'_k, S \setminus U) - f(U, S \setminus U)$ and replace $U$ by $C'_k$ in $S$. Since the added child is unsuccessful, it decreases the diversity of the population but helps to handle plateaus, which turned out to be more effective than restricting the decrease of diversity.

Now, let us analyze the complexity of the genetic algorithm.

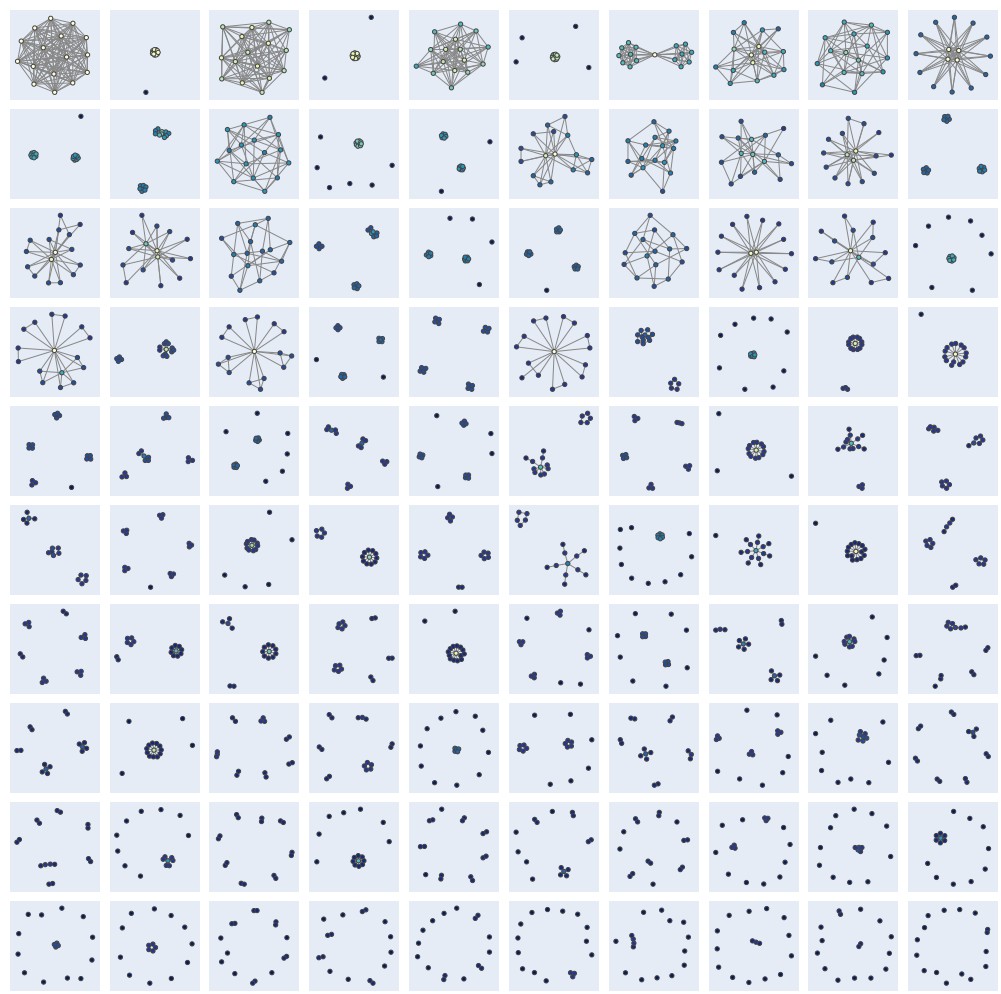

Figure 10: Graphs from IGGM with netLSD-wave: most of the graphs are sparse

**Proposition C.3.** *The time complexity of the genetic algorithm is $O((a + bN)L)$, where $a$ is the complexity of computing a graph representation, $b$ is the complexity of calculating the distance between two representations, and $L$ is the number of update attempts.*

*Proof.* Each step of the genetic algorithm requires $O(N)$ operations for the choice of parents, $O(n^2)$ for the crossover, $O(n)$ for the mutation, where $n$ is the number of nodes in a graph, $O(a + bN)$ to calculate distances between the generated child and every graph in the population, $O(N)$ to find a graph $U$ which gives the largest improvement. So, in total, one step requires $O(n^2 + a + bN)$ operations. Since for all used distances $D$ we have $a \geq n^2$, we get that one step requires $O(a + bN)$ operations. If we run the genetic algorithm for $L$ steps (thus generating exactly $L$ children), the resulting time complexity is $O((a + bN)L)$.

The time complexity of generating the initial population and calculating the fitness for all its elements is small in comparison with $O((a + bN)L)$, so it does not influence the time complexity of the algorithm (given that $L \gg N$). $\qquad\square$

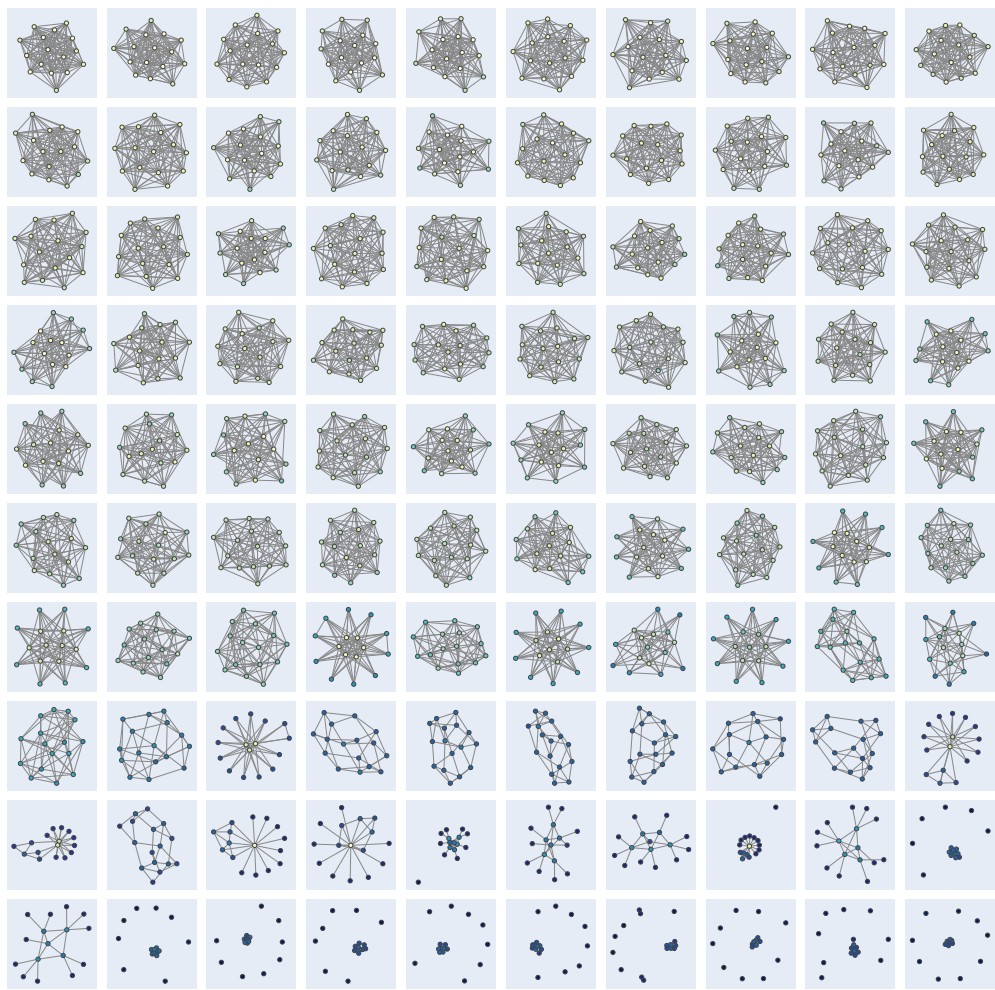

Figure 11: Graphs from IGGM with GCD

### C.4 Local optimization algorithm

**Initial population** The local optimization algorithm supports the same techniques as the genetic algorithm for defining the initial population. Namely, we can pass existing graphs through command line argument or can generate graphs on the fly from random graph models, e.g., ER-0.5 or ER-mix. At the beginning, we compute the fitness $f(G, S \setminus G)$ for every graph $G$ in the population $S$.

Then, we do the following:

1. At each iteration, choose a graph with probability inversely proportional to its fitness[5] and denote this graph by $U$.

2. Pick two distinct nodes $u$ and $v$ from $U$ uniformly at random. If an edge $(u, v)$ exists in $U$, remove it. Otherwise, add $(u, v)$ to the edge list of $U$. Using this atomic operation, we obtain a changed graph $U'$.

3. Compute the fitness $f(U', S \setminus U)$.

4. If $f(U', S \setminus U) > f(U, S \setminus U)$, replace $U$ by $U'$. Otherwise, do nothing and repeat the process.

---

[5]For Energy, we sample with probabilities proportional to $-$fitness.

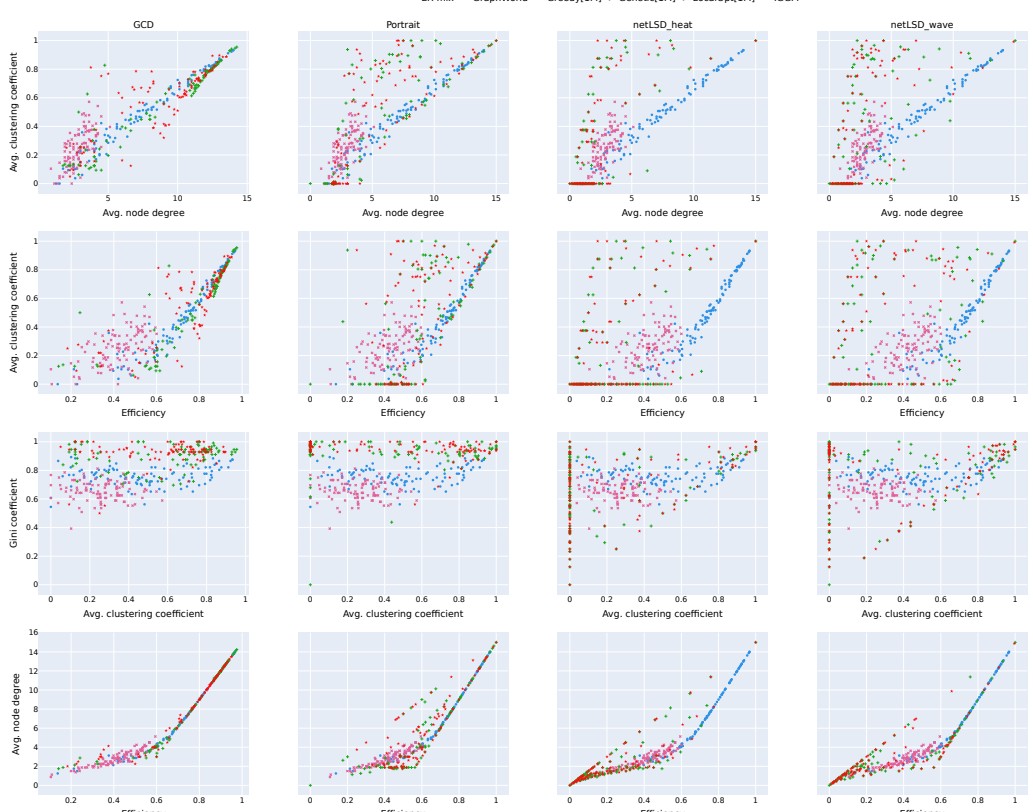

Figure 12: Joint distribution of graph characteristics, extended figure

5. Similarly to the genetic algorithm, we track the previous $K$ attempts and if they fail, we accept the replacement among the last $K$ attempts with the lowest difference between $f(U_k', S \setminus U_k)$ and $f(U_k, S \setminus U_k)$.

6. Repeat the process $L$ times, where $L$ is the total number of attempts.

The following proposition follows from the algorithm description.

**Proposition C.4.** *The complexity of the local optimization algorithm is $O((a + bN)L)$, where $a$ is the complexity of computing a graph representation, $b$ is the complexity of calculating the distance between two representations, and $L$ is the number of update attempts.*

### C.5 Iterative Graph Generative Modeling (IGGM)

In this section, we describe our implementation of the approach based on iterative neural generative modeling for the task of obtaining $N$ structurally diverse graphs.

We denote the number of generated graphs during each generation step by $K$ and the total number of generated graphs by $L$. It is not strictly required but assumed that $L$ is divisible by $K$. Another parameter of the algorithm is $R < K$: $R$ graphs are used to train each graph generative model. In our experiments, we set $R = 10^3$, $K = 10^5$, $L = 10^6$.

To obtain the initial set of graphs to train a generative model on, we do the following: we generate a set of $10R$ graphs from ER-mix and then apply the Greedy algorithm on that set, obtaining $R$ initial graphs, which we denote $S_0$ and start the process.

At each step $t$, we train a generative neural network $g_{\theta_t}$ on the graphs obtained from the previous step $S_{t-1}$; the notation $\theta_t$ represents trainable parameters of the model at step $t$. Then, we use $g_{\theta_t}$ as a fixed random graph generator to create $K \gg R$ graphs and apply the Greedy algorithm to choose $R$ diverse graphs that will form the next training input.

Table 4: Energy optimization results for graphs with 16 nodes

| Setup | GCD | Portrait-div | NetLSD-heat | NetLSD-wave |
|---|---|---|---|---|
| ER-mix | $0.281 \pm 0.0350$ | $43.057 \pm 8.812$ | $72.387 \pm 26.192$ | $0.583 \pm 0.0892$ |
| GraphWorld | $0.466 \pm 0.0151$ | $3.917 \pm 0.0896$ | $5.108 \pm 0.2856$ | $0.621 \pm 0.0117$ |
| Random Graph Generators | $0.553 \pm 0.0167$ | $6.009 \pm 0.2368$ | $116.685 \pm 8.009$ | $1.334 \pm 0.0831$ |
| Greedy[1M] | $0.160 \pm 0.0004$ | $1.287 \pm 0.0029$ | $0.682 \pm 0.0005$ | $0.124 \pm 0.0003$ |
| Greedy[2M] | $0.157 \pm 0.0010$ | $1.278 \pm 0.0033$ | $0.681 \pm 0.0008$ | $0.124 \pm 0.0004$ |
| Greedy[3M] | $0.156 \pm 0.0004$ | $1.274 \pm 0.0018$ | $0.681 \pm 0.0001$ | $0.123 \pm 0.0003$ |
| ER-mix→Genetic[3M] | $0.139 \pm 0.0025$ | $1.264 \pm 0.0031$ | $0.677 \pm 0.0013$ | $\mathbf{0.117 \pm 0.0003}$ |
| Greedy[1M]→Genetic[2M] | $0.139 \pm 0.0018$ | $1.263 \pm 0.0020$ | $0.674 \pm 0.0003$ | $0.118 \pm 0.0005$ |
| Greedy[2M]→Genetic[1M] | $0.141 \pm 0.0012$ | $1.263 \pm 0.0027$ | $0.674 \pm 0.0004$ | $0.118 \pm 0.0005$ |
| ER-mix→Genetic[1M]→LocalOpt[2M] | $0.138 \pm 0.0002$ | $1.259 \pm 0.0007$ | $0.675 \pm 0.0000$ | $\mathbf{0.117 \pm 0.0001}$ |
| Greedy[1M]→LocalOpt[2M] | $0.139 \pm 0.0012$ | $1.255 \pm 0.0006$ | $0.679 \pm 0.0001$ | $0.118 \pm 0.0001$ |
| Greedy[1M] → Genetic[1M] → LocalOpt[1M] | $0.135 \pm 0.0000$ | $1.245 \pm 0.0004$ | $\mathbf{0.673 \pm 0.0001}$ | $\mathbf{0.117 \pm 0.0001}$ |

In our experiments, for $g_{\theta_t}$ we use Discrete Denoising Diffusion model DiGress (Vignac et al., 2023), which is distributed under the MIT License, with its default parameters (including model hyperparameters and training routine) to generate graphs.

As a result, our procedure consists of $\frac{L}{K}$ steps, where at each step $1 \le t \le \frac{L}{K}$ we do:

1. Train generative model $g_{\theta_t}$ on the set of graphs $S_{t-1}$, initial weights are taken from the last weights of the previous iteration;

2. Generate $K$ graphs using $g_{\theta_t}$;

3. Apply the greedy algorithm to the generated set of graphs to obtain $R$ graphs, denote this set by $S_t$.

Note that after each iteration, we can greedily choose $N$ graphs from $S_t$ of size $R$, and thus obtain the set of structurally diverse graphs with the desired size.

The time complexity of the algorithm depends on the number of epochs chosen for each network training step and thus may vary a lot. We used the default parameters of DiGress as a proof of concept.

## D   Experimental setup

**GraphWorld parameters**   For GraphWorld, we vary the model parameters to obtain more diverse graph structures. Namely, we choose the following parameters uniformly: P2Q-ratio from $[1, 10]$, num_communities from $\{1, 2, 3, 4, 5\}$, avg_node_degree form $[4, n-1]$, power_exponent from $[0.5, 1)$.

**Hardware setup**   The experiments have been conducted on the machine with Intel Core i7-7800X @3.50GHz CPU, 2×NVIDIA GeForce RTX 2080 Ti GPUs and 126G RAM. It took us approximately 500 CPU hours to conduct all the experiments.

## E   Additional experiments

**Examples of generated graphs**   Examples of generated graphs are shown in Figures 7-11. We see that when combined with Portrait-div, both Genetic and IGGM generate visually diverse and interesting structures. One can also notice that NetLSD tends to generate many extremely sparse graphs, while GCD generates more dense graphs.

**Visualizing graph characteristics**   Figure 12 visualizes various characteristics of generated graphs for the ER-mix and GraphWorld baselines, IGGM, and the combination of Greedy, Genetic, and LocalOpt. It extends Figure 4 from the main text.

**Extended numerical results**   Table 4 extends the results from Table 1 and also reports the standard deviation based on five independent trials.

Table 5: Energy optimization results for graphs with 64 nodes

| Setup | GCD | Portrait-div | NetLSD-heat | NetLSD-wave |
|---|---|---|---|---|
| ER-mix | 0.400 | 2.236 | 452.885 | 0.454 |
| GraphWorld | 0.442 | 3.746 | 3.509 | 0.510 |
| Random Graph Generators | 0.540 | 5.868 | 112.685 | 1.298 |
| Greedy[1M] | 0.177 | 1.155 | 0.812 | 0.172 |
| Greedy[3M] | 0.175 | 1.148 | 0.796 | 0.169 |
| ER-mix→Genetic[3M] | 0.167 | 1.128 | 0.567 | 0.128 |
| Greedy[1M]→Genetic[2M] | 0.158 | 1.126 | 0.673 | 0.117 |
| ER-mix→Genetic[1M]→LocalOpt[2M] | 0.133 | 1.086 | **0.551** | 0.118 |
| Greedy[1M]→LocalOpt[2M] | 0.132 | 1.082 | 0.603 | 0.117 |
| Greedy[1M] → Genetic[1M] → LocalOpt[1M] | **0.128** | **1.060** | 0.673 | **0.116** |

**Larger graphs** We conducted experiments on larger graphs with $n = 64$ nodes, the results are shown in Table 5 and they are consistent with the results on smaller graphs.

