# OpenReview forum: "Challenges of Generating Structurally Diverse Graphs"
_NeurIPS.cc/2024/Conference — NeurIPS 2024 poster_

### Official Review · Reviewer_VyJM · 2024-07-12

**Soundness:** 2
**Presentation:** 2
**Contribution:** 2
**Rating:** 4
**Confidence:** 4

**Summary:**

This work considers the problem of generating graph topologies that are structurally diverse. The core argument for considering this task is that training and evaluating models for graph learning requires good "coverage" of the space of possible graphs to draw meaningful conclusions. The paper discusses various ways of quantifying the distance between two graphs, as well as ways for measuring diversity for a set of graphs given a distance measure. It then proposes a series of algorithms for generating structurally diverse graphs that essentially start from an initial set of graphs that are altered via local modifications; with changes being accepted if the diversity metric improves. It validates experimentally that the proposed procedures lead to more diverse graphs than, e.g., using Erdos-Renyi graphs or a pre-existing library of graph topologies.

**Strengths:**

The primary strength of the work is that it is very systematic in the way that it approaches the problem. It does a thorough job of reviewing the ways in which graph distances and diversity of a set of graphs can be quantified; and the experiment results indeed achieve the main goal of optimizing structural diversity. The writing quality and clarity of the paper are also good.

**Weaknesses:**

W1. In my opinion, the main weakness is that the work stops short of the main motivating use case of why generating diverse graphs is important. Namely, the work does not carry out an assessment of whether the generated diverse graphs actually improve performance for graph learning tasks (and to what extent; for what types of problems; which distance and diversity metrics are beneficial in practice, etc.). The paper in its current form reads like it is addressing only "half" of the motivating problem.

W2. I am also unsure of the fit to the NeurIPS conference given the above, as the paper does not present findings relevant to machine learning and artificial intelligence in its current form. It may be better suited for a network science or complex networks venue. This would not be the case if the diverse graphs are then used downstream on learning-based tasks.

W3. The proposed methods are fairly brute-force and quite limited in scale (experiments consider graphs with up to $n=64$ nodes, which already seem to require millions of examples). Some form of learning representation is, in my opinion, needed to make these techniques applicable on a larger scale. Furthermore, I would suggest looking into reinforcement learning as a way of carrying out the optimization process by using diversity as a reward signal for goal-directed graph construction or modification (instead of requiring "examples" of diverse networks as with many graph generative models, which is why they are not applicable here as the authors remark).

**Questions:**

Please address W1, W2, W3 above. Additional minor comments:

- First paragraph of the introduction would benefit from citations
- L38: use `\citep` instead
- Another "classic" work relevant to the discussion on encoding structural information based on the graph Laplacian (L123) is [1].
- L175: I'd avoid using $g$ to denote the bijection
- Typos: GraphWold (L313), "graphs graphs" (Fig 3 caption)

[1] Wilson, R. C., Hancock, E. R., & Luo, B. (2005). Pattern vectors from algebraic graph theory. IEEE transactions on pattern analysis and machine intelligence, 27(7), 1112-1124.

**Limitations:**

Limitations are adequately discussed.

---

> ### Author Rebuttal · Authors · 2024-08-07
>
> Thank you for your review! We address the questions and concerns below.
>
> > W1.  In my opinion, the main weakness is that the work stops short of the main motivating use case of why generating diverse graphs is important. Namely, the work does not carry out an assessment of whether the generated diverse graphs actually improve performance for graph learning tasks (and to what extent; for what types of problems; which distance and diversity metrics are beneficial in practice, etc.).
>
> In this work, we introduce the problem of generating diverse graphs and cover several important aspects of this problem: defining diversity, testing several types of algorithms, assessing the performance. While we are motivated by some practical applications, we believe that diving deep into this is way beyond the scope of our work. First, there are plenty of challenges covered in the paper. Second, before evaluating the performance in particular applications, the first step is to generate a reasonably diverse set of graphs that can be used.
>
> We discuss possible applications of diverse graphs in lines 26-39. For one application, namely evaluating neural algorithmic reasoning (NAR) models, we launched some preliminary experiments. We utilized the standard pipeline from the CLRS benchmark [1]. For several algorithmic tasks, we trained a standard NAR model on the train graphs from the benchmark and tested the performance on larger graphs using the following datasets: the standard CLRS test generator (ER-0.5) and the set obtained by the Genetic algorithm with Portrait-Div distance. We get the following performance (average accuracy across 5 different seeds):
>
> | Algorithm | CLSR test | Genetic Portrait-Div |
> |-|:-:|:-:|
> | Bellman-Ford  |  92  |  87 |
> | Dijkstra | 92 |  85 | | Floyd-Warshall |  31 | 29 |
> | MST-Prim | 84 | 83 |
>
> We see that the performance drops in all the cases, implying that the obtained diverse set of graphs is more challenging compared to the baseline test set. This illustrates the importance of testing on as diverse set of graphs as possible.
>
> While these are preliminary results, we speculate that such diverse graphs can also be useful as a part of the training set to improve the robustness and generalizability of neural algorithmic reasoning methods.
>
> However, we believe that investigating particular applications should be the topic of a separate study.
>
> > W2. I am also unsure of the fit to the NeurIPS conference given the above, as the paper does not present findings relevant to machine learning and artificial intelligence in its current form. It may be better suited for a network science or complex networks venue. This would not be the case if the diverse graphs are then used downstream on learning-based tasks.
>
> We respectfully disagree that the subject of the paper does not fit the NeurIPS scope: the topics in the NeurIPS call for papers are quite diverse and include, in particular, theoretical studies that may not directly improve downstream performance.
>
> From the practical perspective, our paper explores new abilities of graph generative approaches. We show how the methods can be adapted to optimize diversity. From the theoretical perspective, we conduct a theoretical analysis of diversity measures which can be of interest by itself.
>
> Also, our problem is motivated by various practical applications and can be of interest to a part of the NeurIPS community. Here, we refer to Review h5vq which characterizes the studied problem to be an important and unsolved problem in graph machine learning.
>
> We are open to further discussions on whether papers of such kind are suitable for NeurIPS.
>
> > W3. The proposed methods are fairly brute-force and quite limited in scale (experiments consider graphs with up to 𝑛=64 nodes, which already seem to require millions of examples).
>
> We expect scalability to be one of the main challenges for future studies. We mention the scalability challenges in lines 370-373 and also make some preliminary experiments with larger graphs (up to 1000 nodes) in Appendix E.3.
>
> Let us mention that for some practical applications, even small graphs can be useful: e.g., in neural algorithmic reasoning, models are usually trained on graphs of size up to 16 nodes, and molecular graphs are also small.
>
> We set our budget to 3M tested graphs as a reasonably large upper bound: algorithms may converge to reasonably diverse sets earlier. However, it is important to note that even such a budget is significantly smaller than the number of non-isomorphic graphs even for $n=16$ (lower bound for this number is $2^{n(n-1)/2} / n! = 2^{120} / 16! > 10^{14}$).
>
> > Some form of learning representation is, in my opinion, needed to make these techniques applicable on a larger scale. Furthermore, I would suggest looking into reinforcement learning as a way of carrying out the optimization process by using diversity as a reward signal for goal-directed graph construction or modification
>
> Thank you for the suggestions! We believe that this can be a good research topic for further studies. Given that our work is pioneering in the field, we did not aim to cover the whole spectrum of possible approaches in the current work.
>
> Thank you for the writing suggestions, we will improve the text accordingly.
>
> [1] P. Veličković et al. The CLRS algorithmic reasoning benchmark. ICML 2022.

---

> > ### Comment · Reviewer_VyJM · 2024-08-12
> >
> > Many thanks for engaging with the points I raised.
> >
> > W1. The scope of the work is not externally defined but wholly under your control. My point is that if the design of study went "deeper" into examining performance on some graph ML tasks instead of "broader" by only looking at how to generate diverse graphs, it would (in my opinion) make for a more compelling set of results. The preliminary experiments do look promising and "close the loop" on your motivating use cases.
> >
> > W2. Just to clarify, this is not a criterion that I am using to penalise the work. My main point is that, in a similar vein to W1, the fit to the conference would be substantially easier to justify if some graph learning tasks were considered. The study is also fairly clearly not a theoretical one.
> >
> > W3. Acknowledging the scalability weakness of the work is a step forward, but nevertheless the authors do not provide ways to address it, or indeed frame the problem in such a way that the techniques would be easy to scale up.
> >
> > I am raising my score to a 4 as a result of the discussion: while the paper has some merits, I still err on the side of rejection due to its weaknesses.

---

> > > ### Author Response · Authors · 2024-08-13
> > >
> > > Thanks a lot for your involvement in the discussion!
> > >
> > > **Regarding W1**, we are happy to hear that you find the experiment promising. We plan to extend this experiment in the revised version as follows: consider the diverse graphs generated based on different graph distances (in our case - Portrait-Div, NetLSD, GCD). Then, we can evaluate various algorithms from the CLRS benchmark on these diverse sets. We expect that diverse sets are more challenging compared to the standard test sets (as shown above for Portrait-Div), but it would also be interesting to see which graph distance leads to more challenging graphs for a particular algorithm. This will give us more insights into graph distances and provide more challenging diverse sets for particular algorithms.
> > >
> > > **Regarding W3**, we agree that it is an important issue, so let us elaborate on this challenge (and we will add this discussion to the revised version).
> > >
> > > When scaling to significantly larger graphs, there are two aspects that one should consider: the computation cost of the chosen graph distance and the speed of convergence of the algorithm.
> > >
> > > Regarding graph distances, their computation cost may vary significantly. For instance, there are simple graph distances based on node degrees which can be easily computed but are less expressive, while some more advanced ones are based on optimal node matching which becomes infeasible even for moderate-size graphs. Among the distances we consider in the paper, the NetLSD implementation turns out to be the fastest. Importantly, most graph distances first compute descriptors of graphs and then measure the distance (e.g., Euclidean) between these descriptors. For better scalability, we suggest using graph distances of this type since in this case we can compute the descriptor for each graph only once and then use it for all distance measurements which significantly reduces the costs.
> > >
> > > Regarding the convergence of the algorithm (in terms of the number of graphs we are allowed to try), this issue is more tricky: the number of possible graph structures grows extremely fast with the number of nodes and thus one cannot hope to explore the whole space via simple bruteforce. Thus, the ability to scale significantly depends on the particular algorithm used.
> > > Let us revisit the algorithms considered in the paper and discuss their potential ability to scale.
> > > - **Greedy**. This algorithm is scalable: assuming that we pre-computed all graph descriptors, the complexity of this algorithm does not depend on the sizes of graphs. Moreover, it is easily parallelizable. However, the success of Greedy critically relies on the initial set. In the paper, we apply this algorithm to a set of graphs generated by diverse generators with different parameters. While this requires a hand-crafted list of models, the solution is very scalable: most random graph generators easily scale to large graphs. If we have a diverse set of graph generators, the greedy algorithm is a very promising model to consider.
> > > - **Local optimization**. This approach is not scalable: it operates with small graph modifications and thus cannot effectively explore the space of all graphs. We find this algorithm suitable for the final tuning of graphs if they are already diverse. A possible way to improve the scalability of LocalOpt is to also allow larger graph modifications, especially in the early steps of the algorithm (e.g., replacing a subgraph instead of changing only one edge).
> > > - **Genetic**. We find this algorithm promising for exploring the space of graphs since its operations (crossovers and mutations) often lead to graphs that significantly differ from the parent graphs. Additionally, we suggest incorporating stochasticity into this algorithm by sampling pairs for fitness calculations instead of evaluating all pairs. This approach could accelerate convergence by allowing more steps within the same timeframe. Finally, in our preliminary experiments on larger graphs, we noticed that Genetic works significantly better when starting with a sufficiently diverse set of graphs.
> > > - **IGGM**. The efficiency of IGGM significantly depends on the efficiency of the underlying generative model. Thus, scaling IGGM to large graphs depends on the future development of efficient GGMs, which lies beyond the scope of this paper. But if we assume that the generative model is sufficiently cheap, IGGM is expected to scale to large graphs, similarly to the greedy algorithm discussed above.
> > >
> > > Overall, we are working on these strategies to enhance the scalability of our framework and appreciate the reviewer's insights on this matter.
> > >
> > > Additionally, we’ve launched more experiments with graphs on 1000 nodes to analyze how scalable are some of the approaches. Due to the limited discussion period, we don't consider larger graphs, but $n = 1000$ is already significantly larger than $n = 16$ used in most of our experiments. We plan to add the results before the end of the discussion period.

---

> ### Author Response · Authors · 2024-08-14
> **Experiments on larger graphs**
>
> As promised above, we conducted some additional experiments for graphs on 1000 nodes. Here, when we optimize diversity, we use either GCD or NetLSD as a graph distance (as Portrait-Div is significantly slower). We analyze Genetic starting from GraphWorld. Since sufficiently diverse starting set is important for faster convergence, for GraphWorld we vary all its parameters to increase diversity of the starting sample. The obtained learning curves can be found in our code repository in `/figures` folder. These curves show how the diversity measure improves with iterations. Several conclusion can be made. First, diversity significantly increases. Second, one can see that the algorithms converge relatively fast, especially for the NetLSD graph distance. This confirms that Genetic has a good potential for scalability.

---

### Official Review · Reviewer_wU7k · 2024-07-12

**Soundness:** 2
**Presentation:** 3
**Contribution:** 2
**Rating:** 5
**Confidence:** 3

**Summary:**

The authors investigate the problem of generating graphs that are structurally diverse. Specifically, the graphs should be as different from each other as possible in terms of their properties.
Towards this, the authors first propose a new way to measure graph diversity based on the idea of energy(combined with several graph distances).
They describe several algorithms for generating diverse graphs, namely greedy, genetic based and  local optimization, and neural -based.

The authors mention that in order to test many graph algorithms or models, it  requires diverse graphs. Otherwise, results might be misleading, only reflecting the tested graph types.

**Strengths:**

1. The paper propose an interesting problem of generating structurally diverse graphs and its importance in different fields such as evaluating graph neural networks and their expressive power etc.

2. The paper is easy to read.

3. The authors define an interesting idea of energy based functions for measuring diversity.

4. Empirical analysis shows better performance in comparison to baselines.

**Weaknesses:**

1. It is not clear to me whether there is a data distribution which is learned.

2. If in  (1)  there is no data, how does one apply the proposed idea to generate graph that follow a distribution? On what data will we evaluated GNN on?

3. What is the usage of diversity without fidelity metrics: such as structural similarity between 2 datasets( training and generated)?

Eg : see https://arxiv.org/abs/2001.08184
GraphGen: A Scalable Approach to Domain-agnostic Labeled Graph Generation.
See diversity and fidelity section.

**Questions:**

Please see weakness.

I am mostly concerned on the application of the proposed work. How does one use it?

Is it possible to model some data distribution+ with diversity?

**Limitations:**

Yes

---

> ### Author Rebuttal · Authors · 2024-08-07
>
> Thank you for your review! We address the questions and concerns below.
>
> > 1. It is not clear to me whether there is a data distribution which is learned.
>
> One of the main differences between our setup and previous works on graph generation is that we do not have any data distribution that we want to mimic. Our goal is to generate a set of graphs that are maximally diverse.
>
> If we define diversity in terms of a particular graph distance then, roughly speaking, one may think that there is a latent space related to this distance measure and we want to learn a uniform distribution over this space. Our proposed non-generative algorithms (e.g., Genetic and LocalOpt) do not sample from this space but iteratively explore it with different techniques. Our generative-based algorithm IGGM aims to learn the uniform distribution mentioned above. To train the IGGM model, we start with some sufficiently diverse distribution, then learn it with a graph generative model, use this model to obtain a more diverse set of graphs, and repeat the procedure.
>
> > 2. If in (1) there is no data, how does one apply the proposed idea to generate graph that follow a distribution? On what data will we evaluated GNN on?
>
> As mentioned above, we do not mimic a given distribution, we aim at generating a maximally diverse set of graphs. After we define what we mean by “diversity of a set of graphs”, the problem is well-defined and does not require any training set. Could you please clarify your question so we can properly address it?
>
> > 3. What is the usage of diversity without fidelity metrics: such as structural similarity between 2 datasets( training and generated)?*
>
> As explained above, we do not have training data, so there is no possible comparison of structural similarity between training and generated datasets.
>
> > I am mostly concerned on the application of the proposed work. How does one use it?
>
> The main question of the work is how one can generate a set of graphs that are maximally diverse. As we mention in the introduction, we foresee the following applications of the obtained graphs:
> - analyzing the performance of a graph algorithm;
> - estimating how well a heuristic algorithm approximates the true solution for a graph problem;
> - evaluating neural approximations of graph algorithms;
> - training neural approximations of graph algorithms to improve their robustness and generalizability;
> - evaluating graph neural networks and their expressive power.
>
> In all these cases, algorithms and models should be tested on as diverse graph instances as possible since otherwise the results can be biased towards particular properties of the test set.
>
> > Is it possible to model some data distribution+ with diversity?
>
> Yes, one can potentially combine a measure of diversity with similarity to a given distribution in one loss function and then use any of the proposed algorithms to optimize this.
>
> We hope that our responses address the raised questions and we are open to further discussions.

---

> > ### Comment · Reviewer_wU7k · 2024-08-10
> > **Thanks**
> >
> > I thank the authors for their answer. I increase my score.

---

### Official Review · Reviewer_h5vQ · 2024-07-13

**Soundness:** 4
**Presentation:** 4
**Contribution:** 4
**Rating:** 7
**Confidence:** 5

**Summary:**

This paper focuses on the problem of generating structurally diverse graphs, an important problem for evaluating graph algorithms.  In short, if you are trying to generate a set of graphs to evaluate an algorithm's performance (runtime or otherwise) one desires a set of graphs that "span the space" of possible graph types in order to make robust estimates of the performance.

The paper first covers different measures of graph similarity, and then goes into 'energy' based algorithms for structure diversity optimization.   Its here that the work may lose a little novelty (its more about the problem and less about the solution).  However all-in-all, I think its a nice contribution that highlights an important (unsolved) problem in graph machine learning.

**Strengths:**

+ topic is important and understudied
+ excellent presentation
+ interesting experimental results

**Weaknesses:**

- the problem is still unsolved
- since there's not much work, there aren't obvious benchmarks to beat
- I have some doubts about the scalability (and therefore practicality) of the presented solution
- the work could probably use more connections to the diversity sampling literature, since once one has a similarity metric, it seems like this is just diversity sampling

**Questions:**

1.  It seems like the entire question rests on the graph distance measure used, and you avoid this discussing this.  What's the best choice?  Alternatively: how could one tell what the best choice for them would be?

2.  How does the work from the diversity sampling literature relate to this work (in particular the optimizations in section 3.)  Section 2.3 addresses two recent related works, but seems there are probably a lot of relevant citations here -- there's a whole cottage industry on diversity sampling.

**Limitations:**

Yes

---

> ### Author Rebuttal · Authors · 2024-08-07
>
> Thank you for your constructive comments and positive feedback! Let us address the weaknesses and questions.
>
> > the problem is still unsolved
>
> We agree with this and we are not expecting this problem to be completely solved in the near future. The problem is challenging, starting from the difficulties with defining diversity for a set of graphs. We hope that our work will stimulate further discussions and progress on this problem. However, we believe that we made an important step towards solving the problem: the proposed methods indeed generate quite interesting and diverse graphs, as can be seen in Figure 3.
>
> > since there's not much work, there aren't obvious benchmarks to beat
>
> Indeed, it is hard to find proper baselines for pioneering studies. Hence, the main goal of our research is to formulate the problem and see how effective are different types of approaches. However, we do our best to design strong baselines based on existing random graph generators and the GraphWorld benchmark.
>
> > I have some doubts about the scalability (and therefore practicality) of the presented solution
>
> We expect scalability to be one of the main challenges for future studies. We mention the scalability challenges in lines 370-373 and also make some preliminary experiments with larger graphs (up to 1000 nodes) in Appendix E.3. We do not dive deeper into scalability issues since there are plenty of challenges even for small graphs. However, we do agree that the scalability question is important. Let us also mention that for some practical applications, even such small graphs can be useful: e.g., in neural algorithmic reasoning, models are usually trained on graphs of size up to 16 nodes, and molecular graphs are also small.
>
> > It seems like the entire question rests on the graph distance measure used, and you avoid this discussing this. What's the best choice? Alternatively: how could one tell what the best choice for them would be?
>
> The problem of the best graph distance measure has been studied for a long time, and there is no clear answer yet, as all distances tend to have biases toward some graph characteristics. It is an important and long-standing problem, so we do not aim to answer this within our work and choose several representative distances for our empirical study (to show that our approaches can be applied to any chosen graph distance).
>
> However, we do discuss and compare graph distances in our experiments. In Appendix E.4 (Figures 12, 13), we show that depending on a chosen graph distance, the obtained graphs may significantly differ. By generating diverse graphs using a particular distance measure and then inspecting the properties of the obtained graphs, one can get new insights about what structural properties this distance measure is sensitive to. For instance, we noticed that NetLSD is biased towards sparse graphs and thus most of the obtained diverse graphs are sparse.
>
> We believe that the best choice of a distance measure may depend on the application. For instance, in the molecular domain, substructures of a molecule may heavily determine its properties, and thus GCD can be a good option since it is sensitive to small substructure counts. On the other hand, if diverse graphs are needed for testing some graph algorithms, then diversity in terms of graph diameter or average shortest path length can be desirable, and in this case Portrait Divergence can be considered as a favorable choice.
>
> > How does the work from the diversity sampling literature relate to this work (in particular the optimizations in section 3.) Section 2.3 addresses two recent related works, but seems there are probably a lot of relevant citations here -- there's a whole cottage industry on diversity sampling.
>
> Thanks for pointing this out, we will mention diversity sampling in the text. Diversity sampling refers to techniques used to select a subset of datapoints from a larger dataset that are representative of the dataset's diversity. Unfortunately, we cannot directly apply these methods to our task since diversity sampling does not go beyond the existing dataset: it is assumed that the dataset to sample from is given and one can iterate through its elements. In contrast, our goal is to generate new graphs: we cannot iterate through all non-isomorphic graphs on $n$ nodes. Section 3.1 of our paper (Greedy baseline) is the one most related to diversity sampling: in this section, we pre-generate an initial set of graphs and then choose the most diverse graphs from this set.
>
> As far as we know, there is not much research on measures of diversity in diversity sampling literature. For instance, diversity is often defined as the sum of pairwise distances (our Average) or its modifications [1,2]. In other cases, diversity can be defined in terms of a variety of class labels or other sample features [3]. In contrast, graphs can vary in structure, connectivity, and many other properties and thus we cannot resort to such simple diversity measures in the more complex graph domain.
>
> [1] S. Agarwal et al. Contextual diversity for active learning. ECCV 2020.
>
> [2] Y. Yang et al. Multi-class active learning by uncertainty sampling with diversity maximization. IJCV 2015.
>
> [3] Y. Geifman et al. Deep active learning over the long tail. 2017.
>
> We are open to further discussions!

---

> ### Comment · Reviewer_h5vQ · 2024-08-13
>
> Thanks to the authors for addressing my comments.  I am raising my score.

---

### Official Review · Reviewer_myRm · 2024-07-13

**Soundness:** 3
**Presentation:** 2
**Contribution:** 2
**Rating:** 5
**Confidence:** 2

**Summary:**

This paper goes through the diversity of graphs and proposes the relevant generation process for diverse graphs. Various generation methods, such as genetic algorithms and greedy algorithms (based on diverse random graph generators), are studied and theoretical results are provided to guarantee the lower bound from the diversity respective.

**Strengths:**

1. This paper is well motivated, and the generation of diverse graphs is a good domain for research.

2. This paper is well written and the idea is easy to follow.

3. The experiment results are sound.

**Weaknesses:**

1. The novelty of this paper should be better stressed, especially for the framework of measuring the diversity via energy.

2. The choice of the measuring of graph diversity is questionable. Overall, the matching of graphs is  combinatorics, popular metrics such as graph edit distance to measure their similarity are fundamental NP-hard problem in graph theory. I think this part should also better discussed.

**Questions:**

N.A.

**Limitations:**

Please refer to Limitations.

---

> ### Author Rebuttal · Authors · 2024-08-07
>
> Thank you for your review and positive assessment of our work! We address the weaknesses below.
>
> > W1. The novelty of this paper should be better stressed, especially for the framework of measuring the diversity via energy.
>
> To the best of our knowledge, the problem of generating structurally diverse graphs hasn’t been studied before and we make the first step in this direction. Standard machine learning algorithms cannot be directly applied to this problem and thus we had to adapt them to the task at hand. Regarding diversity evaluation, we prove that typically used measures have critical drawbacks and show that energy is more suitable for this task. Our theoretical analysis is also novel.
>
> > W2. The choice of the measuring of graph diversity is questionable. Overall, the matching of graphs is combinatorics, popular metrics such as graph edit distance to measure their similarity are fundamental NP-hard problem in graph theory. I think this part should also better discussed.
>
> We discuss the problem of comparing two graphs in Section 2.2 and Appendix A. Here, we briefly discuss measures based on the optimal node matching in lines 111-115. We don't consider such measures in our experiments since they are computationally expensive. We will add graph edit distance to this discussion, thanks for the suggestion.
>
> We hope that these modifications address your concerns. If you have any more concerns or ideas about what should be specified in the text, please let us know.

---

### Author Response · Authors · 2024-08-14
**Post-discussion summary**

We would like to thank all the reviewers for the productive discussion and positive feedback! This discussion helped us to improve the paper.

Following the suggestions, we’ve extended our discussion of scalability, added more experiments with larger graphs, and added some experiments that illustrate practical applications of diverse graphs.

We would like to conclude with some of our thoughts regarding the problem addressed in our work. The task of generating diverse graphs turns out to be a challenging problem. The challenges start with defining diversity for a set of graphs since even defining pairwise graph distance is a long-standing unsolved problem. Then, one needs to develop algorithms for the task since standard approaches cannot be directly applied. In our paper, we made the first steps to addressing these challenges. We are personally very encouraged by the obtained results. In particular, Figure 3 in the paper (and Figures 7-8 in Appendix) show that the proposed approaches allow one to generate visually nice structurally diverse graphs even when starting from simple Erdős–Rényi graphs. We hope that our work will stimulate further research in this field.

---

### Decision · Program_Chairs · 2024-09-25

**Decision:**

Accept (poster)

**Comment:**

The paper addresses an important problem, i.e. generating structurally diverse graph to make evaluation of graph-based ML methods more robust, for the first time (as far as I know). While the proposed solution has some flaws (e.g. not very computationally appealing), it is a first step in addressing these challenges and it presents encouraging results. I recommend acceptance.

**Note to SAC** I will add a summary of the paper once the final decision has been taken.